# Consistency of the $k_n$-nearest neighbor rule under adaptive sampling

**Robi Bhattacharjee**[1]      **Sanjoy Dasgupta**[2]      **Geelon So**[2]

[1]University of Tübingen and Tübingen AI Center
[2]Department of Computer Science and Engineering, UC San Diego

## Abstract

In the *adaptive sampling* model of online learning, future prediction tasks can be arbitrarily dependent on the past. Every round, an adversary selects an instance to test the learner. After the learner makes a prediction, a noisy label is drawn from an underlying conditional label distribution and is revealed to both learner and adversary. A learner is consistent if it eventually performs no worse than the Bayes predictor. We study the $k_n$-nearest neighbor learner within this setting. In the worst-case, the learner will fail because an adaptive process can generate spurious patterns out of noise. However, under the mild smoothing assumption that the process generating the instances is *uniformly absolutely continuous* and that choice of $(k_n)_n$ is reasonable, the $k_n$-nearest neighbor rule is online consistent.

## 1   Introduction

We study *binary classification with noisy labels* in the online setting where predictions are made using the $k_n$-nearest neighbor rule (Fix and Hodges, 1951; Cover and Hart, 1967). Let $(\mathcal{X}, \rho)$ be a instance space equipped with a metric, let $\mathcal{Y} = \{0, 1\}$ be a binary label space, and let the labels be chosen by nature, drawn from a conditional label distribution defined by $\eta : \mathcal{X} \to [0, 1]$,

$$\eta(x) \equiv \Pr(Y = 1 | X = x).$$

In the *adaptive sampling model*, the label noise is benign, but the sequence of prediction tasks can be adversarial and adapt to the sequence of observed labels. For $n = 1, 2, \ldots,$

- a data-generating process with knowledge of the past selects an instance $X_n \in \mathcal{X}$,
- the learner makes a prediction $\hat{Y}_n \in \mathcal{Y}$,
- nature reveals a label $Y_n$, freshly drawn from the Bernoulli distribution, $\mathrm{Ber}(\eta(X_n))$.

The goal is to make as few mistakes $\hat{Y}_n \neq Y_n$ as possible.

If the learner knew $\eta$, then it should predict the Bayes optimal label $Y_n^* = \mathbb{1}\{\eta(X_n) \geq 1/2\}$, as this minimizes the expected error at each time step. But this strategy is not generally possible when $\eta$ is unknown. Still, we measure our learner against it: a learner is *consistent* if its asymptotic mistake rate is no worse than what is achieved by making the Bayes optimal prediction every round:

$$\limsup_{N \to \infty} \frac{1}{N} \sum_{n=1}^{N} \mathbb{1}\{\hat{Y}_n \neq Y_n\} - \mathbb{1}\{Y_n^* \neq Y_n\} \leq 0 \qquad \text{a.s.} \tag{1}$$

In other words, the learner is asymptotically consistent if its performance in the long run is on par with the best predictor given knowledge of ground-truth label distribution.

39th Conference on Neural Information Processing Systems (NeurIPS 2025).

---
**Algorithm 1** The $k_n$-nearest neighbor rule
___
1: **for** $n = 1, 2, \ldots$ **do**
2:      Receive the instance $X_n$
3:      Predict the majority vote label of the $k_n$ nearest neighbors $X_n^{(1)}, \ldots, X_n^{(k_n)}$,

$$\hat{Y}_n = \mathbb{1} \left\{ \frac{1}{k_n} \sum_{k=1}^{k_n} Y_n^{(k)} \geq 1/2 \right\}$$

4:      Observe and memorize the label $Y_n$
5: **end for**
___

The learner we consider is the $k_n$-*nearest neighbor rule*. It memorizes all data it sees. To predict on the $n$th instance $X_n$, it sorts the $n - 1$ previously memorized data points by distance to $X_n$,

$$X_n^{(1)}, \ldots, X_n^{(n-1)}, \qquad \text{where} \quad \rho\big(X_n, X_n^{(k)}\big) \leq \rho\big(X_n, X_n^{(k+1)}\big),$$

and it predicts with the majority vote over the labels of the $k_n$ nearest neighbors, as shown in Algorithm 1. If there are distance ties, then we let the data point that arrived first take precedence. Many other, but not all, tie-breaking mechanisms are reasonable, but let us leave those details to Appendix E. While this is not needed, we assume that distance ties almost never occur.

## 1.1 Consistency of $k_n$-nearest neighbor under non-adaptive sampling

For appropriate sequences $(k_n)_n$, the $k_n$-nearest neighbor rule is consistent generally when the instances are not adaptive to the labels (Chaudhuri and Dasgupta, 2014, and references therein). The instances are usually assumed to be generated by an i.i.d. process, but a more general assumption suffices (Kulkarni and Posner, 1995), where each label $Y_i$ is conditionally independent of given $X_i$,

$$\Pr(Y_i | \mathbb{X}, \mathbb{Y}_{-i}) = \Pr(Y_i | X_i).$$

Here, $\mathbb{X} = (X_1, X_2, \ldots)$ is the sequence of instances and $\mathbb{Y}_{-i} = (Y_1, \ldots, Y_{i-1}, Y_{i+1}, \ldots)$ is the sequence of labels without $Y_i$. Note that this precludes adaptive sampling mechanisms, where the selection of downstream instances can depend on the realization of $Y_i$. But under this non-adaptive setting, the proof of consistency is conceptually straightforward. Say that $\mathcal{X}$ is a sufficiently nice metric space and $\eta$ is continuous. There are two key ideas:

(a) If $k_n = \omega(\log n)$ grows sufficiently fast, then the law of large numbers will always be in effect, so that the empirical conditional means converge to their conditional expectations:

$$\frac{1}{k_n} \sum_{k=1}^{k_n} Y_n^{(k)} \rightarrow \frac{1}{k_n} \sum_{k=1}^{k_n} \eta(X_n^{(k)}) \qquad \text{(statistical convergence)}$$

(b) If $k_n = o(n)$ does not grow too fast, then the $k_n$ nearest neighbors $X_n^{(k)}$ converge to $X_n$. By the continuity of $\eta$, the conditional means over the neighbors tend to converge:

$$\frac{1}{k_n} \sum_{k=1}^{k_n} \eta(X_n^{(k)}) \rightarrow \eta(X_n) \qquad \text{(geometric convergence)}$$

By chaining these two limiting behaviors together, we obtain a very informal proof that the empirical label frequencies over the $k_n$ nearest neighbors eventually converge to $\eta(X_n)$. And at this point, the learner's prediction becomes consistent with the Bayes optimal predictor.

To extend consistency beyond the non-adaptive setting, we will show that both types of statistical and geometric convergence can be achieved under much weaker assumptions.

## 1.2 Main results

We first show that in the worst-case setting, the $k_n$-nearest neighbor rule can fail to be consistent. However, at least in our counter-example, an adversary really needs to select points carefully in

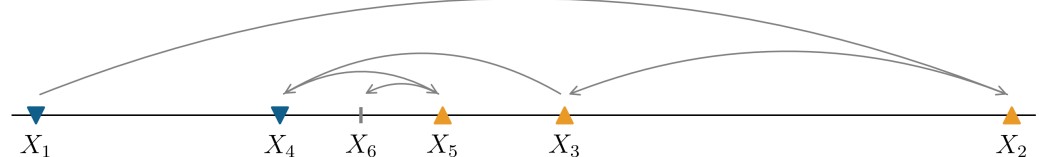

Figure 1: The *binary search* adaptive sampling strategy efficiently finds a good threshold when data on the line is labeled by a threshold function. However, when the labels are generated by i.i.d. noise, the binary search strategy also generates a dataset that is linearly separable (see Example 6). For example, here the existing data $X_1, \ldots, X_5$ is separable. By querying a point like $X_6$ between the two classes, the data will continue to be separable no matter which label is realized.

order to construct such a worst-case sequence. This raises the question: how brittle are such hard constructions? In a recent work, Dasgupta and So (2024) provided an answer for the adaptive but noiseless setting, showing that the 1-nearest neighbor rule is consistent under very mild conditions on the data-generating process. That is, worst-case sequences almost never occur.

This work considers the same question, but in the presence of noise. Is the $k_n$-nearest neighbor rule also generally a viable prediction strategy in adaptive settings? Or, does noise pose a much harder challenge to learning? To answer these questions, we impose the same, mild condition on the data process introduced by Dasgupta and So (2024). It ensures that a process selects from small regions with small probability, generalizing the *smoothness* condition of Haghtalab et al. (2020):

**Definition 1** (Uniform absolute continuity). A data process is *uniformly dominated* by $\upsilon$ if for any $\varepsilon > 0$, there exists $\delta > 0$ such that when a measurable set $A \subset \mathcal{X}$ satisfies $\upsilon(A) < \delta$, then:

$$\forall n, \quad \Pr\left(X_n \in A \,\middle|\, \mathbb{X}_{<n}, \mathbb{Y}_{<n}\right) < \varepsilon.$$

We say that $\mathbb{X} = (X_n)_n$ is *uniformly absolutely continuous* with respect to $\upsilon$ at rate $\varepsilon(\delta)$. We say that the process is *smoothed* or *L-dominated* if $\varepsilon(\delta) \leq L\delta$ for some constant $L \geq 1$.

For $\varepsilon(\delta)$-uniformly dominated processes, we show that the $k_n$-nearest neighbor rule is consistent when the conditional label distribution $\eta$ is continuous and when $k_n$ satisfies for some $c > 0$:

$$k_n = \omega\left(\log \frac{1}{\varepsilon^{-1}(n^{-(1+c)})}\right) \qquad \text{and} \qquad k_n = o(n). \tag{2}$$

In the case when the data process is $L$-dominated, we recover the standard condition on $k_n$ that is imposed to obtain consistency in the i.i.d. setting, which is that $k_n = \omega(\log n)$ and $k_n = o(n)$.

**Theorem 2** (Consistency of the $k_n$-nearest neighbor rule). *Let $(\mathcal{X}, \rho, \upsilon)$ be a separable metric space with a finite Borel measure. Let $\eta : \mathcal{X} \to [0,1]$ be continuous. Let $\mathbb{X}$ be uniformly dominated at rate $\varepsilon(\delta)$. If $k_n$ satisfies (2), then the $k_n$-nearest neighbor rule is consistent almost surely.*

While Theorem 2 is already very general, it requires that $\eta$ is continuous. The next result greatly relaxes this condition to admit all measurable $\eta$. To achieve this, we impose stronger conditions on the space. The general class of *upper doubling spaces* was studied by Dasgupta and So (2024), where they showed that 1-NN is consistent for any measurable label function $\eta$ in the realizable setting:

**Definition 3** (Upper doubling). A metric space $(\mathcal{X}, \rho)$ is *doubling* with doubling dimension $d$ if every ball $B(x, r)$ can be covered by $2^d$ balls of radii $r/2$. A $d$-doubling space with measure $\upsilon$ is *upper doubling* if there exists $c > 0$ such that for all $B(x, r)$, we have $\upsilon\big(B(x, r)\big) \leq cr^d$.

We show a corresponding result in such spaces for $k_n$-nearest neighbor rule in the noisy, adaptive sampling setting. In fact, it further allows us to relax the condition on $(k_n)_n$. Theorem 2 required the sequence of $k_n$ to satisfy (2), which depends on the rate function, $\varepsilon(\delta)$. In upper doubling spaces, the following regularity condition suffices, which encompasses most choices a practitioner might make, as any sequence of the form $k_n = n^\alpha (\log n)^\beta$ suffices if $0 < \alpha < 1$ or if $\alpha = 0$ and $\beta > 1$.

**Definition 4** (Regular sequence). A non-decreasing sequence $(k_n)_n$ is said to be *regular* if grows between $k_n = \omega(\log n)$ and $k_n = o(n/\log n)$, and if for all $c \in \mathbb{N}$, it satisfies $\lim_{n \to \infty} k_{cn}/k_n \to 1$.

**Theorem 5** (Universal consistency in upper doubling spaces). *Let $(k_n)_n$ be a regular sequence, and $(\mathcal{X}, \rho, \upsilon)$ be an upper doubling space. Let $\mathbb{X}$ be uniformly dominated by $\upsilon$ and $\eta : \mathcal{X} \to [0,1]$ be measurable. Then the $k_n$-nearest neighbor rule is online consistent with respect to $(\mathbb{X}, \eta)$.*

## 1.3 Related work

Nearest neighbor methods are fundamental to non-parametric learning, where for the most part, they are studied in settings with strong statistical independence assumptions (Fix and Hodges, 1951; Cover and Hart, 1967; Stone, 1977; Devroye et al., 1994, 2013; Chaudhuri and Dasgupta, 2014; Hanneke et al., 2020; Györfi and Weiss, 2021). Expanding beyond i.i.d. or stationary processes, Kulkarni and Posner (1995) remove independence assumptions across instances, but still assume that sampling is non-adaptive (and thus independent of the labels). This work forgoes making explicit independence assumptions across instances or labels, but rather imposes a 'bounded precision' constraint on the data-generating process that was introduced by Dasgupta and So (2024).

This paper contributes to the *smoothed* or *non-worst-case* analysis of learning, which studies learning settings that are in between i.i.d. and worst-case (Rakhlin et al., 2011; Haghtalab et al., 2020, 2024; Hanneke, 2021; Block et al., 2022, 2024; Blanchard and Jaillet, 2023; Dasgupta and So, 2024). Many of the work in smoothed online learning take place in the parametric setting. Our work provides complementary results for the non-parametric setting. Moreover, we provide some initial directions for extending the theory of sequential uniform convergence to the smoothed adaptive setting, taking a related but distinct approach to Rakhlin et al. (2015).

## 2 Learning on noisy and adaptively-sampled data

To illustrate the challenges of learning on noisy sequential data, we'll first describe an example of a data-generating mechanism that produces data that looks starkly different from what is 'expected': it turns out that an adaptive sampling strategy can make patterns out of random noise (Figure 1).

In the following, an instance sequence is generated via the binary search sampling strategy, while the labels are independent flips of the same coin. As an artifact of the sampling procedure, the data appears to be linearly separable, even though the underlying labeling mechanism is uniform throughout space. Thus, the observable pattern in this case will fail to generalize to future data.

**Example 6** (Binary search on noise). *Fix $p \in (0,1)$. Let $\mathcal{X} = [0,1]$ and let $\eta(x) = p$ be constant. That is, each round, regardless of $X_n$, the label $Y_n \sim \mathrm{Ber}(p)$ is generated by a coin flip with bias $p$. Construct $(X_1, Y_1, \ldots, X_N, Y_N)$ as follows. Initialize $(X_0, Y_0) = (0,0)$. For $n = 1, \ldots, N$,*

- *select $X_n = X_{n-1} + \frac{1}{2^n} \cdot (-1)^{Y_{n-1}}$,*

- *draw $Y_n \sim \mathrm{Ber}(p)$.*

*At time $N$, data points left of $X_N$ have the label 0, while those to the right have the label 1.*

This idea can be used to construct an adversarial sequence for the $k_n$-nearest neighbor rule. For example, if $p > 1/2$, the Bayes optimal rule predicts 1 everywhere. But, the $k_n$-nearest neighbor rule will predict the suboptimal label 0 for a long time on instances queried to the left of $X_N$.

**Proposition 7** (Inconsistency of $k_n$-NN). *Let $\mathcal{X} = [0,1]$ and let labels on $\mathcal{X}$ be generated by a Bernoulli distribution $\mathrm{Ber}(p)$ where $p \in (1/2, 1)$. Let $(k_n)_n$ be regular. There is an adaptive sampling strategy that generates a data stream for which the $k_n$-nearest neighbor rule is not consistent:*

$$\limsup_{N \to \infty} \frac{1}{N} \sum_{n=1}^{N} \mathbb{1}\{\hat{Y}_n \neq Y_n^*\} \geq \frac{1-p}{8} \qquad \text{a.s.}$$

To a statistician trained in the i.i.d. setting, these examples where 'statistical convergence' fail can be counterintuitive. In Example 6, specifically what may be surprising here is the large discrepancy between *empirical* and *expected* label frequencies. Let $I$ be the open interval $(X_0, X_N)$. Then:

$$\frac{1}{|I|} \sum_{X_i \in I} Y_i = 0 \qquad \text{and} \qquad \frac{1}{|I|} \sum_{X_i \in I} \mathbb{E}[Y_i | X_i] = p.$$

This is true even though $I$ contains a very large number of points with high probability, when $p$ is bounded away from 0 or 1. At first, this seems to violate the martingale law of large numbers.

Here's a false proof of convergence: if we carelessly apply Azuma-Hoeffding's inequality to the bounded, mean-zero random variables $Y_i - \mathbb{E}[Y_i | X_i]$ taken in sequence from the interval $I$, we would

conclude that the deviation between the empirical and expected label frequency must tend to zero. The problem is that the set $I$ itself carries information about each $Y_i$, so the appropriate martingale difference sequence over which to apply Azuma-Hoeffding's needs to also condition on $I$. Now, as $I$ and $Y_i$ are dependent, the conditional expectation of $Y_i$ given $I$ is not $p$. Actually, in this case, $X_N$ encodes the first $N-1$ data points $(X_1, Y_1, \ldots, X_{N-1}, Y_{N-1})$ exactly, and so $\mathbb{E}[Y_i | I] = Y_i$.

Still, this does not satisfactorily resolve the difference between the i.i.d. and sequential setting. When $\mathbb{X}$ is an i.i.d. process, we can even search for a corresponding worst interval $I'$ with the largest discrepancy between the empirical and expected frequencies; $I'$ is quite dependent on $\mathbb{Y}$. But, because the class of intervals on the line has finite VC dimension, so long as $I'$ contains sufficiently many points, we are guaranteed that the empirical and expected label frequencies will be close (for example, see Theorem 5 of Balsubramani et al. (2019)). This follows from i.i.d. uniform convergence theory, which shows that with high probability, the empirical and expected label frequencies simultaneously converge for *all* intervals. For a more formal comparison, see Proposition A.1.

It can be unsettling that statistical intuition from the i.i.d. setting does not seem to transfer to the sequential setting. Many processes, such as scientific discovery, take place under the adaptive sampling model: we can think of instances as a sequence of experiments that a scientific community performs and labels as the corresponding outcomes. This data-generating process is certainly not i.i.d., as earlier results inform which experiments are performed next. And so, it is important to understand where the new failure modes that arise with adaptivity come from, and how likely are they to occur. To do so, let's take a detour into the *conditional mean estimation* problem.

## 3 Online conditional mean estimation

Let $(X_1, Y_1, \ldots, X_N, Y_N)$ be an adaptively sampled dataset, following the model defined at the start. Even in the sequential setting, the discrepancy between the empirical and expected frequencies is well-understood by standard martingale concentration; by Azuma-Hoeffding's inequality,

$$\Pr\left( \left| \underbrace{\frac{1}{N} \sum_{n=1}^{N} Y_i}_{\text{empirical frequency}} - \underbrace{\frac{1}{N} \sum_{n=1}^{N} \eta(X_i)}_{\text{expected frequency}} \right| \geq t \right) \leq 2\exp(-2Nt^2).$$

But now, suppose that a data analyst would like to make finer-grained inferences beyond estimating the average label over the whole dataset. In particular, to define the *conditional mean estimation* problem, we say that a *query* is any subset of indices $Q \subset [N]$, possibly chosen with knowledge of the whole dataset. The data analyst would like to use the *conditional empirical frequency* $\hat{\eta}_N(Q)$ to estimate the *conditional expected frequency* $\bar{\eta}_N(Q)$,[1]

$$\underbrace{\hat{\eta}_N(Q) := \frac{1}{|Q|} \sum_{i \in Q} Y_i}_{\text{conditional empirical frequency}} \qquad \text{and} \qquad \underbrace{\bar{\eta}_N(Q) := \frac{1}{|Q|} \sum_{i \in Q} \eta(X_i)}_{\text{conditional expected frequency}}.$$

We can think of *statistically valid* queries as those that have strong concentration guarantees showing that these two quantities converge quickly to each other as $|Q|$ grows.

Of particular interest to this paper is the *spatial query*: for any region $A$ and capacity $k \in [N]$, it selects for the first $k$ instances that land in a fixed region $A$.

**Definition 8** (Spatial query). Let $A$ be measurable and $k \in [N]$. Let $(X_1, Y_1, \ldots, X_N, Y_N)$ be an adaptively sampled dataset. The *spatial query* $Q_{N,k}(A)$ is the query at time $N$:

$$Q_{N,k}(A) = \big\{ n \in [N] : X_n \in A \text{ and } |A \cap \{X_1, \ldots, X_n\}| \leq k \big\}.$$

When $\mathcal{A}$ is a class of measurable sets, let $\mathcal{Q}_{N,k}(\mathcal{A}) = \{Q_{N,k}(A) : A \in \mathcal{A}\}$.

Spatial queries are easily seen to a type of *label-oblivious* query, which turn out to be statistically valid. Label-oblivious queries can be sequentially constructed, but the decision to insert $n$ into the query must be made before $Y_n$ is revealed. Thus, we can apply Azuma-Hoeffding's to the martingale difference sequence $Y_n - \mathbb{E}[Y_n | X_1, Y_1, \ldots, X_n]$ to obtain the concentration result in Lemma 10.

---

[1]As a technicality, if $|Q| = 0$, we let $\hat{\eta}_N(Q) = \bar{\eta}_N(Q) = 0$.

**Definition 9** (Label-oblivious query). Let $(X_1, Y_1, \ldots, X_N, Y_N)$ be an adaptively sampled dataset. At time $N$, a query $Q \subset [N]$ is *label-oblivious* if for all $n \in [N]$ the conditional independence holds:

$$\mathbb{1}\{n \in Q\} \perp\!\!\!\perp Y_n \mid X_1, Y_1, \ldots, X_n.$$

Thus, the decision to include $n \in Q$ can be made after $X_n$ is revealed, but before $Y_n$ is realized.

The following provides a simple concentration bound. Notice that there is a factor of $k$ that does not appear in the original Azuma-Hoeffding's inequality. This is because the sample size $|Q|$ is adaptive, and so we take a naive union bound over all possible sample sizes. Tighter bounds are possible (Balsubramani, 2014; Zhao et al., 2016), but this is good enough for us.

**Lemma 10** (Concentration for label-oblivious queries). *Suppose that $(X_1, Y_1, \ldots, X_N, Y_N)$ is an adaptively sampled dataset and $Q$ is a label-oblivious query such that $|Q| \leq k$ almost surely. Then:*

$$\Pr\left(\left|\hat{\eta}_N(Q) - \bar{\eta}_N(Q)\right| \geq t\right) \leq 2k \exp\left(-2|Q|t^2\right).$$

Besides label-oblivious queries, we would also like to study queries that may be somewhat *label-dependent*. Such queries arise in exploratory or adaptive data analysis, where the question being asked can depend on the observed data. Of course, a good statistical sensibility tells us that the query cannot be overly sensitive to the dataset itself; for example, a query that cherry picks all instances with the label 1 should certainly be ruled out. But, some others seem reasonable, at least from an i.i.d. standpoint, like the $k$-nearest neighbor query for a fixed $x$. After all, if there are no ties, then for every realization of the $k$-nearest neighbor query, there is a spatial query selecting the same instances:

**Definition 11** ($k$-nearest neighbor ball query). Fix $x \in \mathcal{X}$ and $k \in [N]$. Let $(X_1, Y_1, \ldots, X_N, Y_N)$ be an adaptively sampled dataset. Then:

- Let $\mathcal{B}_x = \{\bar{B}(x, r) : r \geq 0\}$ consist of closed balls centered at $x$. A *ball query* at $x$ is any:

$$Q_{N,k}(\bar{B}) \qquad \text{where} \quad \bar{B} \in \mathcal{B}_x,$$

  so that $Q_{N,k}(\bar{B})$ selects for the first $k$ instances $X_n$ that land in $\bar{B}$.

- The *$k$-nearest neighbor ball query* at $x$ is the adaptive query:

$$Q_{N,k}(x) = Q_{N,k}\left(\bar{B}(x, r)\right) \qquad \text{where} \quad r = \arg\min_{s > 0} \left\{\left|Q_{N,k}\left(\bar{B}(x, s)\right)\right| \geq k\right\},$$

  so $Q_{N,k}(x)$ selects for instances in the smallest ball $\bar{B}(x, r)$ that contains at least $k$ instances.

However, as Proposition 7 shows, even such a seemingly benign form of label-dependency can still over-fit to past data in the worst-case setting. The reason that the $k$-nearest neighbor query is valid in the i.i.d. setting is because uniform convergence holds over the class of balls $\mathcal{B}_x$, which is to say that all ball queries are *simultaneously* valid: no matter how dependent the ball query is on the dataset, convergence is still guaranteed with high probability. As Proposition A.1 shows, we cannot generally expect uniform convergence to hold for $\mathcal{B}_x$ in the worst-case. However, we show now that these worst-case processes are in a sense very rare; under mild constraints, they never occur.

### 3.1 Smoothed uniform convergence: concentration for adaptive spatial queries

Let $\mathcal{X}$ have a Borel probability measure $\upsilon$ and let $\mathcal{A}$ be a class of measurable sets. In this section, we provide a basic uniform convergence result for spatial queries over the class $\mathcal{A}$ assuming that the adaptive sampling mechanism is uniform dominated by $\upsilon$.

The idea to prove uniform convergence is simple: suppose that we can approximate $\mathcal{A}$ by a finite collection of sets $\mathcal{C}$, which we will call a *sandwiching cover*. To obtain uniform convergence of $\mathcal{A}$, we show that (a) uniform convergence holds on $\mathcal{C}$, and that (b) the approximation error achieved by $\mathcal{C}$ is small, which is possible to show using the uniform domination condition. To key to controlling this approximation error is the *sandwiching* property:

**Definition 12** (Sandwiching cover). Let $(\mathcal{X}, \upsilon)$ be a measure space and let $\mathcal{A}$ be a collection of measurable sets. Let $\alpha \geq 0$. An *$\alpha$-sandwiching cover* of $\mathcal{A}$ is a collection $\mathcal{C}$ of measurable sets such that for all $A \in \mathcal{A}$, there exist $A_{\text{in}}, A_{\text{out}} \in \mathcal{C}$ such that:

$$A_{\text{in}} \subset A \subset A_{\text{out}} \qquad \text{and} \qquad \upsilon(A_{\text{out}} \setminus A_{\text{in}}) \leq \alpha.$$

Let $\mathcal{N}_{\mathcal{A}}(\alpha)$ be the *$\alpha$-sandwich number* of $\mathcal{A}$, the size of the smallest $\alpha$-sandwiching cover of $\mathcal{A}$.

The sandwich number for balls centered at a point will be particularly relevant:

**Lemma 13** (Sandwich number for balls centered at $x$). *Let $(\mathcal{X}, \rho, \upsilon)$ be a separable metric space with a Borel probability measure. Fix $x \in \mathcal{X}$ and let $\mathcal{B}_x$ be the set of closed balls centered at $x$. Then, for any $\alpha \in [0, 1]$, the $\alpha$-sandwich number $\mathcal{N}_{\mathcal{B}_x}(\alpha)$ of $\mathcal{B}_x$ is at most $4/\alpha$.*

**Lemma 14** (Uniform concentration for spatial queries). *Let $(X_1, Y_1, \ldots, X_N, Y_N)$ be an adaptively sampled dataset on $(\mathcal{X}, \rho, \upsilon)$, a metric measure space where $\upsilon$ is a Borel probability measure. Let $\mathcal{A}$ be a family of measurable sets and let $\mathcal{Q}_{N,k}(\mathcal{A})$ be a corresponding class of spatial queries. Suppose that $\mathbb{X}$ is $\varepsilon(\delta)$-uniformly dominated. For any $p \in (0, 1)$, with probability at least $1 - p$,*

$$\forall Q \in \mathcal{Q}_{N,k}(\mathcal{A}), \qquad \left| \hat{\eta}_N(Q) - \bar{\eta}_N(Q) \right| \leq \frac{2\ell}{|Q|} + \sqrt{\frac{1}{2(|Q| - \ell)} \log \frac{k \cdot \mathcal{N}_{\mathcal{A}}(\alpha)}{p}},$$

*whenever $\alpha$ satisfies $\varepsilon(\alpha) = 1/N$ and $\ell \geq \max\{2 \log \frac{2\mathcal{N}_{\mathcal{A}}(\alpha)}{p}, e^2\}$.*

*Proof sketch.* Let $\mathcal{C}$ be a minimal $\alpha$-sandwiching cover of $\mathcal{A}$, so that $|\mathcal{C}| \leq \mathcal{N}_{\mathcal{A}}(\alpha)$. Union bounding over all queries $Q \in \mathcal{Q}_{N,k}(\mathcal{C})$, we obtain from Lemma 10 that with probability at least $1 - p/2$,

$$\forall Q \in \mathcal{Q}_{N,k}(\mathcal{C}), \qquad \left| \hat{\eta}_N(Q) - \bar{\eta}_N(Q) \right| \leq \sqrt{\frac{1}{2|Q|} \log \frac{k \cdot \mathcal{N}_{\mathcal{A}}(\alpha)}{p}}. \tag{3}$$

To extend this large-deviation bound to the rest of $\mathcal{Q}_{N,k}(\mathcal{A})$, we shall use the fact that every $A \in \mathcal{A}$ is sandwiched between two elements of $\mathcal{C}$ that are $\alpha$-close $A_{\text{in}} \subset A \subset A_{\text{out}}$. We just need to ensure that the region $A_{\text{out}} \setminus A_{\text{out}}$ does not contain a very large number of points from $\mathbb{X}_{\leq N}$. There are at most $\mathcal{N}_{\mathcal{A}}(\alpha)^2$ such difference regions, so it is possible to union bound over them as well: Lemma B.1 shows that none of these regions contains more than $\ell$ points. The contribution of these points are accounted for by the $2\ell/|Q|$ term and the slight adjustment to $1/(|Q| - \ell)$. $\qquad \square$

We instantiate this lemma for the $k$-nearest neighbor query in Corollary 2 of Appendix B.3.

# 4 Consistency of $k_n$-nearest neighbor rule for continuous $\eta$

Here, we will work under the assumption that almost surely no tie-breaking is needed for example, when the instance space is a Euclidean space with a measure $\upsilon$ that is absolutely continuous with respect to the Lebesgue measure. Theorem 2, where ties can exist, is proved in Appendix E.

**Theorem 15** (Consistency of the $k_n$-nearest neighbor rule). *Let $(\mathcal{X}, \rho, \upsilon)$ be a separable metric space with a finite Borel measure. Let $\eta : \mathcal{X} \to [0, 1]$ be continuous. Suppose that $\mathbb{X}$ is uniformly dominated at rate $\varepsilon(\delta)$ and that almost surely there are no distance ties. If $k_n$ satisfies (2), then the $k_n$-nearest neighbor rule is consistent almost surely.*

*Proof sketch.* Under uniform domination, statistical and geometric convergence hold (Propositions 16 and 18). Then, apply triangle inequality. The proof is in Appendix C and is slightly more subtle. $\quad \square$

## 4.1 Statistical convergence of $k_n$-nearest neighbor

In the introduction, we described the *statistical convergence* of the $k_n$-nearest neighbor query in the i.i.d. setting: the empirical label frequencies converge to the conditional expected frequencies,

$$\frac{1}{k_n} \sum_{k=1}^{k_n} Y_n^{(k)} \to \frac{1}{k_n} \sum_{k=1}^{k_n} \eta(X_n^{(k)}),$$

informally speaking. In the previous section, Lemma 14 and Corollary 2 prove concentration for $k_n$-nearest neighbor queries for instances $Z \sim \upsilon$ drawn independently of an adaptively-generated data set. We use this to show statistical convergence for uniformly dominated process.

**Proposition 16** (Statistical convergence of $k_n$-nearest neighbor). *Let $(\mathcal{X}, \rho, \upsilon)$ be a metric space with a Borel probability measure. Let $\eta : \mathcal{X} \to [0, 1]$ be arbitrary. Suppose that when the sampling*

*process is uniformly dominated, no distance ties occur, almost surely. Let $(X_1, Y_1, \ldots)$ be adaptively sampled by an $\varepsilon(\delta)$-uniformly dominated process and let $k_n$ satisfy (2). Then:*

$$\lim_{n \to \infty} \left| \frac{1}{k_n} \sum_{k=1}^{k_n} Y_n^{(k)} - \frac{1}{k_n} \sum_{k=1}^{k_n} \eta(X_n^{(k)}) \right| = 0 \qquad \text{a.s.}$$

To prove this, we extend Corollary 2 by a partial coupling. Consider two parallel mechanisms:

1. Let $(X_1, Y_1, \ldots, X_N, Y_N, X_{N+1})$ be the adaptively sampled process of interest.
2. Let $(X_1, Y_1, \ldots, X_N, Y_N, Z)$ be a process that coincides with the first, until the last draw from $\mathcal{X}$, at which point an independent draw $Z \sim \upsilon$ is sampled instead.

We would like to bound the chance that a $k$-nearest neighbor query centered at $X_{N+1}$ is statistically non-convergent. In general, this seems quite challenging, since we do not have much control over $X_{N+1}$ except that it is generated by a uniformly dominated process. However, we are able to bound the corresponding event for $Z$. The following lemma relates the probabilities of these two events:

**Lemma 17** (Partial coupling bound). *Let $\mathcal{D}$ below be the outcome of an adaptive sampling process that is $\varepsilon(\delta)$-uniformly dominated by a probability measure $\upsilon$; and, let $\mathcal{D}'$ be the outcome of an alternate mechanism that replace the last instance by an independent draw $Z \sim \upsilon$:*

$$\mathcal{D} = (X_1, Y_1, \ldots, X_N, Y_N, X_{N+1}) \qquad and \qquad \mathcal{D}' = (X_1, Y_1, \ldots, X_N, Y_N, Z).$$

*Let $\mathcal{F}$ be the $\sigma$-algebra adapted to this sequence and let $E$ be any $\mathcal{F}$-measurable event. Then:*

$$\Pr_{\mathcal{D}}(E) \leq \inf_{s > 0} \varepsilon(s) + \frac{1}{s} \Pr_{\mathcal{D}'}(E).$$

*Proof.* Let $\mathcal{G} = \sigma(X_1, Y_1, \ldots, X_N, Y_N)$ be the $\sigma$-algebra adapted to the first $N$ labeled data points. As $E$ is $\mathcal{F}$-measurable, the random set $A = \{x \in \mathcal{X} : (X_1, \ldots, Y_N, x) \in E\}$, is $\mathcal{G}$-measurable. This is the set of outcomes conditioned on $X_1, \ldots, Y_N$ for which $E$ happens. For any $s > 0$, we obtain:

$$\Pr_{\mathcal{D}}(E) \overset{(i)}{=} \mathop{\mathbb{E}}_{X_1, \ldots, Y_N} \left[ \mathbb{E} \left[ \mathbb{1}\{Z \in A\} \,\middle|\, \mathcal{G} \right] \right]$$

$$\overset{(ii)}{\leq} \mathop{\mathbb{E}}_{X_1, \ldots, Y_N} \left[ \varepsilon(\upsilon(A)) \right]$$

$$\overset{(iii)}{\leq} \varepsilon(s) + \Pr\left( \upsilon(A) > s \right) \overset{(iv)}{\leq} \varepsilon(s) + \frac{1}{s} \Pr_{\mathcal{D}'}(E),$$

applying (i) the law of total expectations, (ii) uniform domination, (iii) the upper bound on $\varepsilon(\upsilon(A))$ by $\varepsilon(s) + \mathbb{1}\{\upsilon(A) > s\}$, and (iv) Markov's inequality. Optimizing over $s > 0$ yields the result. $\square$

*Proof of Proposition 16.* Fix any $s > 0$ and time $n$, define the random variable:

$$\mathcal{D} = (X_1, Y_1, \ldots, X_{n-1}, Y_{n-1}, X_n),$$

which are generated by an $\varepsilon(\delta)$-uniformly dominated adaptive sampling process. Define $E_{n,t}$ as the event that the empirical and expected conditional frequencies at time $n$ have $t$-large discrepancy:

$$E_{n,t} = \left\{ \left| \frac{1}{k_n} \sum_{k=1}^{k_n} Y_n^{(k)} - \frac{1}{k_n} \sum_{k=1}^{k_n} \eta(X_n^{(k)}) \right| \geq t \right\}.$$

By assumption, $k_n = \omega\left( \log \frac{1}{\varepsilon^{-1}(n^{-(1+c)})} \right)$. We show that:

$$\Pr\left( E_{n,t} \right) = o\left( n^{-(1+c)} \right).$$

As the sum of these probability converges, the Borel-Cantelli lemma (Lemma B.3) implies that the discrepancy exceeds $t$ finitely often, yielding:

$$\limsup_{n \to \infty} \left| \frac{1}{k_n} \sum_{k=1}^{k_n} Y_n^{(k)} - \frac{1}{k_n} \sum_{k=1}^{k_n} \eta(X_n^{(k)}) \right| \leq t \qquad \text{a.s.}$$

Letting $t$ go to zero gives the result.

Instead of bounding $\Pr(E_{n,t})$ directly, we consider a parallel process:
$$\mathcal{D}' = (X_1, Y_1, \ldots, X_{n-1}, Y_{n-1}, Z),$$
where the first $n-1$ labeled data points are generated by the same adaptive sampling process, but where the last instance $Z$ is independently drawn from $\upsilon$. Lemma 17 shows that for any $s > 0$:
$$\Pr_{\mathcal{D}}(E_{n,t}) \leq \varepsilon(s) + \frac{1}{s}\Pr_{\mathcal{D}'}(E_{n,t}),$$
and so it suffices to show that eventually, we can set $s$ so that:
$$s = \varepsilon^{-1}\left(n^{-(1+c)}\right) \qquad \text{and} \qquad \Pr_{\mathcal{D}'}(E) \leq s^2,$$
since $\varepsilon(\delta)$ is lower bounded by $\delta$. As there are no distance ties, this follows by our choice of $k_n$ and the concentration result for the $k_n$-nearest neighbor query, Corollary 2, in which we let $p = s^2$. $\qquad \square$

## 4.2 Geometric convergence of $k_n$-nearest neighbor

The next result shows that when the process is uniformly dominated, then $X_n^{(1)}, \ldots, X_n^{(k_n)}$ have conditional label frequencies that converge to that of $X_n$ in the following sense.

**Proposition 18** (Geometric convergence of $k_n$-nearest neighbor). *Let $(\mathcal{X}, \rho, \upsilon)$ be a space with a separable metric $\rho$ and a finite, Borel measure $\upsilon$. Let $\eta : \mathcal{X} \to [0, 1]$ be continuous. Suppose that $\mathbb{X}$ is uniformly dominated at rate $\varepsilon(\delta)$. If $k_n = o(n)$, then for any $s > 0$:*
$$\limsup_{N \to \infty} \frac{1}{N} \sum_{n=1}^{N} \mathbb{1}\left\{\left|\frac{1}{k_n}\sum_{k=1}^{k_n}\eta(X_n^{(k)}) - \eta(X_n)\right| \geq s\right\} = 0 \qquad \text{a.s.}$$

*Proof.* Fix $s > 0$ and let $\mathcal{B}$ be a countable open cover of $\mathcal{X}$ by balls $B = B(z, r)$ with the property:
$$\sup_{x, x' \in B(z, 3r)} |\eta(x) - \eta(x')| < s.$$
Such a cover exists by the continuity of $\eta$ and the separability of $\mathcal{X}$. Now, define $E_n$ to be the event:
$$E_n = \left\{\text{there is a ball } B \in \mathcal{B} \text{ such that } X_n \in B \text{ and } |B \cap \mathbb{X}_{<n}| \geq k_n\right\}.$$
Lemma B.6 shows that when $E_n$ occurs, all $k_n$-nearest neighbors of $X_n$ must also be close, and so the labels are also $s$-close:
$$E_n \subset \left\{\max_{1 \leq k \leq k_n} |\eta(X_n) - \eta(X_n^{(k)})| \leq s\right\}.$$
Therefore, to prove the result, it suffices to show that for any $\varepsilon > 0$ that:
$$\limsup_{N \to \infty} \frac{1}{N} \sum_{n=1}^{N} \mathbb{1}\{E_n \text{ does not occur}\} \leq \varepsilon \qquad \text{a.s.}$$

To do so, fix $\delta > 0$ and take a finite subcover $\mathcal{B}'$ of $\mathcal{B}$ of size $M$ that covers all but a $\delta$-fraction of $\mathcal{X}$. Denote the remaining uncovered region by $\mathcal{X}_\delta = \mathcal{X} \setminus \bigcup \mathcal{B}'$. Then, we decompose the event:
$$\mathbb{1}\{E_n \text{ does not occur}\} \leq \mathbb{1}\{X_n \in \mathcal{X}_\delta\} + \mathbb{1}\{X_n \notin \mathcal{X}_\delta \text{ and } E_n \text{ does not occur}\}.$$
By the uniform absolute continuity of $\mathbb{X}$, only an $\varepsilon$-fraction of points can land in the remainder:
$$\limsup_{N \to \infty} \frac{1}{N} \sum_{n=1}^{N} \mathbb{1}\{X_n \in \mathcal{X}_\delta\} \leq \varepsilon(\delta) \qquad \text{a.s.,} \tag{4}$$
by Lemma B.5. We also have that at any time $n$, at most $k_n M$ points can land in a ball $B \in \mathcal{B}'$ containing fewer than $k_n$ points. Since $k_n M / n \to 0$, this contributes nothing to the asymptotic rate:
$$\limsup_{N \to \infty} \frac{1}{N} \sum_{n=1}^{N} \mathbb{1}\{X_n \notin \mathcal{X}_\delta \text{ and } E_n \text{ does not occur}\} = 0 \qquad \text{a.s.} \tag{5}$$
The result follows from setting $\delta$ sufficiently small and summing Equations 4 and 5. This proof shows that the geometric convergence of $k_n$-nearest neighbor also holds under a weaker condition implied by Lemma B.5 called *ergodic continuity* (Definition B.4), introduced in Dasgupta and So (2024). $\qquad \square$

# 5 Universal consistency on upper doubling spaces

In this section, we introduce the key technical innovations for Theorem 5, which shows universal consistency of the $k_n$-nearest neighbor rule in upper doubling spaces. The first idea is to show that there are (random) subsequences of $\mathbb{X}$ that are well-behaved, on which Theorem 2 would imply consistency. These sequences appear to be generated by an $L$-dominated process where $\varepsilon(\delta) \leq L\delta$, and the labels seem to be sampled from a continuous label distribution $\eta_0 : \mathcal{X} \to [0, 1]$. Moreover, $L$ and $\eta_0$ can be chosen so that these subsequences only fail to capture an arbitrarily small fraction of total instances. In the following, we let $\mathbb{I}$ be an *indicator process*, which is simply a binary process we use to indicate instances in the subsequences $\mathbb{X}[\mathbb{I}] \equiv \{X_n : I_n = 1\}$ and $\mathbb{X}[1 - \mathbb{I}] \equiv \{X_n : I_n = 0\}$.

**Definition 19** (Indicator process). An *indicator process* $\mathbb{I} = (I_n)_n$ is a $\{0, 1\}$-valued stochastic process. Given another stochastic process $\mathbb{X}$, we say that $\mathbb{I}$ is *adapted* to $\mathbb{X}$ if $\mathbb{I}$ is adapted to the natural filtration of $\mathbb{X}$. We say that $\mathbb{I}$ is *asymptotically rate-limited* by $\gamma > 0$ if:

$$\limsup_{N \to \infty} \frac{1}{N} \sum_{n=1}^{N} I_n \leq \gamma \quad \text{a.s.}$$

**Lemma 20** (Reduction to Lipschitz setting). *Let $\upsilon$ be a probability measure on $\mathcal{X}$, and $\eta : \mathcal{X} \to [0, 1]$ be measurable. Let $\mathbb{X}$ be uniformly dominated by $\upsilon$. For any $0 < \gamma < 1/2$, there exists $L > 0$, a continuous map $\eta_0 : \mathcal{X} \to [0, 1]$, and an indicator process $\mathbb{I}$ adapted to $\mathbb{X}$ such that $\mathbb{I}$ is asymptotically $\gamma$-rate-limited, the subsequence $\mathbb{X}[1 - \mathbb{I}]$ is $L$-dominated by $\upsilon$, and $\eta(\mathbb{X}[1 - \mathbb{I}]) = \eta_0(\mathbb{X}[1 - \mathbb{I}])$.*

However, this by itself is not enough to guarantee universal consistency: points in $\mathbb{X}[\mathbb{I}]$ that are not well-behaved may have undue influence on the predictions if they are often a $k_n$-nearest neighbor of downstream instances. In order to control this, the next lemma shows that points in $\mathbb{X}[\mathbb{I}]$ have limited impact within the set of $k_n$-nearest neighbors, as long as $\mathcal{X}$ is upper doubling. In particular, if a significant fraction of the nearest neighbors do come from $\mathbb{X}[\mathbb{I}]$, then we would discover them through subsampling. For analysis, we define the following triangular array of indicator variables $\mathbb{J} := (J_{m,n})_{m \leq n}$, whose randomness is completely independent of $\mathbb{X}$ and $\mathbb{Y}$,

$$\forall m \leq n, \qquad J_{m,n} \sim \text{Ber}(1/k_n). \tag{6}$$

**Lemma 21** (Long-term influence bound). *Let $(\mathcal{X}, \rho, \upsilon)$ be an upper doubling space and $(k_n)_n$ a regular sequence. There exist $c_1, c_2 > 0$ so that the following holds. Let $\mathbb{X}$ be uniformly dominated at rate $\varepsilon(\delta)$ and $\mathbb{I}$ be an indicator process adapted to $\mathbb{X}$ asymptotically rate-limited by $\gamma > 0$, and $\mathbb{J}$ be given by (6). For any $\delta > 0$, the rate that an $\mathbb{I}$-indicated $k_n$-nearest neighbor is sampled by $\mathbb{J}$ is:*

$$\limsup_{N \to \infty} \frac{1}{N} \sum_{n=1}^{N} \mathbb{1}\left( \exists X_m \in \{X_n^{(1)}, \ldots, X_n^{(k_n)}\} : I_m J_{m,n} = 1 \right) < \gamma\left( c_1 + c_2 \log \frac{1}{\delta} \right) + \varepsilon(\delta) \quad \text{a.s.}$$

*Proof sketch of Theorem 5.* For any fixed $\gamma > 0$, construct $\mathbb{I}$ and $\eta_0$ via Lemma 20, so that $\eta_0$ is continuous and $\mathbb{I}$ is $\gamma$-rated limited. Since we can choose $\gamma$ to be arbitrarily small, we may ignore mistakes made during indicated times $I_n = 1$. Instead, we focus on bounding mistakes when $I_n = 0$. On these times, the instance $X_n$ lands in the region where $\eta$ is equal to $\eta_0$. And so, applying the same argument used for Theorem 15, we obtain statistical and geometric convergence:

$$\limsup_{N \to \infty} \frac{1}{N} \sum_{n=1}^{N} \left| \frac{1}{k_n} \sum_{k=1}^{k_n} Y_n^{(k)} - \frac{1}{k_n} \sum_{k=1}^{k_n} \eta(X_n^{(k)}) \right| + \left| \frac{1}{k_n} \sum_{k=1}^{k_n} \eta_0(X_n^{(k)}) - \eta_0(X_n) \right| = 0.$$

But this time, statistical convergence is in terms of $\eta$, and geometric convergence in terms of $\eta_0$, so we cannot apply triangle inequality yet. We also need that the discrepancy between $\eta$ and $\eta_0$ when averaged over sets of $k_n$-nearest neighbors can also be made to be arbitrarily small most of the time.

Notice that whenever the discrepancy is larger than a constant, then a constant fraction of $k_n$-nearest neighbors must be indicated by $\mathbb{I}$. Thus, if we sample from the nearest neighbors using $\mathbb{J}$ on such an event, we are likely to detect at least one indicated neighbor. However, Lemma 21 shows that the asymptotic rate of detecting indicated nearest neighbors can be made arbitrarily small: set $\delta = \gamma$ and let $\gamma$ become vanishingly small. And so, the rate at which this discrepancy is larger than any fixed constant is negligible. This allows us to complete the triangle inequality, proving universal consistency of the $k_n$-nearest neighbor rule in upper doubling spaces under uniform domination. □

The formal proofs for this section are in Appendix D.

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

# A Proofs for Section 2

**Proposition 7** (Inconsistency of $k_n$-NN). *Let $\mathcal{X} = [0,1]$ and let labels on $\mathcal{X}$ be generated by a Bernoulli distribution $\mathrm{Ber}(p)$ where $p \in (1/2, 1)$. Let $(k_n)_n$ be regular. There is an adaptive sampling strategy that generates a data stream for which the $k_n$-nearest neighbor rule is not consistent:*

$$\limsup_{N \to \infty} \frac{1}{N} \sum_{n=1}^{N} \mathbb{1}\{\hat{Y}_n \neq Y_n^*\} \geq \frac{1-p}{8} \qquad \text{a.s.}$$

*Proof.* Define the $L$th epoch to be the set of times in $\mathbb{N}$,

$$\mathrm{Epoch}_L = \{2^{L+1}, \dots, 2^{L+2} - 1\},$$

so that the $L$th epoch contains $2^{L+1}$ time stamps. Define $R_L$ to be the rate at which the learner made a Bayes inconsistent prediction during the $L$th epoch:

$$R_L = \frac{1}{2^{L+1}} \sum_{n \in \mathrm{Epoch}_L} \mathbb{1}\{\hat{Y}_n \neq Y_n^*\}.$$

Notice that as the lengths of each epoch doubles each time, we have by the end of the $L$th epoch:

$$\frac{1}{2^{L+2} - 1} \sum_{n=1}^{2^{L+2}-1} \mathbb{1}\{\hat{Y}_n \neq Y_n^*\} \geq \frac{1}{2} R_L.$$

Even if in all the previous epochs, no mistakes were made, that would only reduce the inconsistency rate by half. And so, it suffices to prove that:

$$\limsup_{N \to \infty} R_L \geq \frac{1-p}{4} \qquad \text{a.s.}$$

To do so, we define the following adaptive sampling strategy, which restarts every epoch:

- **Initialize:** select a closed interval $[a,b] \subset \mathcal{X}$ that contains no previous data points, and by a change of coordinates, renormalize it to $[0,1]$.

- **For the first half of the epoch:** run the binary search sampling strategy for $2^L$ rounds on the renormalized interval. At the end, sort the data from this epoch by the usual ordering:

$$X^{(1)} < \cdots < X^{(2^L)}.$$

  By construction, the first $M$ sorted instances have label 0, where:

$$\mathbb{E}[M] = (1-p)2^L.$$

- **For the second half of the epoch:** exploit the region containing many 0's. Define:

$$k^* = \max_{n \in \mathrm{Epoch}_L} k_n \qquad \text{and} \qquad k_* = \min n \in \mathrm{Epoch}_L \, k_n.$$

  Let $x$ be a point sandwiched between two segments of data points of size $k^*$ all labeled 0,

$$\underbrace{X^{(i-k^*+1)} < \cdots < X^{(i)}}_{\text{segment of } k^* \text{ data points}} < x < \underbrace{X^{(i+1)} < \cdots < X^{(i+k^*)}}_{\text{segment of } k^* \text{ data points}},$$

  then by sampling the point $x$ consecutively for $k_*/2$ times, we can induce the $k_n$-nearest neighbor rule to predict 0 every single time. As there are at least $\lfloor (M - k^*)/k^* \rfloor$ such sandwiched points, we can induce at least:

$$\left( \frac{M}{k^*} - 2 \right) \cdot \frac{k_*}{2} \qquad \text{inconsistencies.}$$

  Therefore, in the $L$th epoch, in expectation, the rate of inconsistency is at least:

$$\mathbb{E}[R_L] \geq \frac{1}{2^{L+1}} \left( \frac{(1-p)2^L - 2k^*}{k^*} \right) \cdot \frac{k_*}{2}.$$

Using the regularity of $k_n$, we have that $\lim_{L\to\infty} k_*/k^* = 1$, so that over all the epochs:

$$\limsup_{L\to\infty} \mathbb{E}\left[R_L\right] \geq \frac{(1-p)}{4}.$$

Thus, by apply the martingale law of large numbers on $(R_L)_L$, we obtain:

$$\limsup_{L\to\infty} R_L \geq \frac{(1-p)}{4} \quad \text{a.s.},$$

which implies the result. $\qquad\square$

**Proposition A.1** (Intervals with large deviation). *Two datasets on $\mathcal{X} = [0,1]$ and $\mathcal{Y} = \{0,1\}$ are generated as follows. The labels have conditional distribution $\eta(x) = 1/2$ everywhere.*

- *Let $(X_1, Y_1, \ldots, X_N, Y_N)$ be adaptively sampled by binary search, as defined in Example 1.*

- *Let $(X_1', Y_1', \ldots, X_N', Y_N')$ be sampled by a uniform i.i.d. process on $\mathcal{X} \times \mathcal{Y}$.*

*Let $I = (X_a, X_b)$ be an interval with endpoints chosen uniformly at random from the adaptive dataset. For any $\varepsilon > 0$, the empirical frequency and expected frequency are unlikely to be similar:*

$$\Pr\left(\left|\sum_{X_n \in I} Y_n - \frac{|I|}{2}\right| \leq N\varepsilon\right) \leq 2\varepsilon. \qquad \text{(non-convergence in adaptive setting)}$$

*Let $I' = (X_a', X_b')$ be an interval with endpoints chosen uniformly at random from the i.i.d. dataset. For any $\varepsilon > 0$, the empirical frequency and expected frequency are unlikely to have large deviation:*

$$\Pr\left(\left|\sum_{X_n' \in I'} Y_n' - \frac{|I'|}{2}\right| \geq N\varepsilon\right) \leq 2\exp(-2N\varepsilon^2). \qquad \text{(convergence in i.i.d. setting)}$$

*Proof.* 1. Non-convergence in the adaptive setting. Sort the dataset generated by Example 6, so that:
$$X^{(1)} < \cdots < X^{(N)}.$$

By construction, the labels are also sorted; there is a threshold $k \in \{0, 1, \ldots, N\}$ where:

$$Y^{(i)} = \mathbb{1}\left\{i > k + \frac{1}{2}\right\}.$$

Let $I = (X^{(i)}, X^{(j)})$ be an interval. The discrepancy is therefore given by:

$$\left|\sum_{X_n \in I} Y_n - \frac{|I|}{2}\right| = \left|\frac{i+j}{2} - \left(k + \frac{1}{2}\right)\right|$$

It follows that an interval has discrepancy less than $N\varepsilon$ if and only if:

$$\frac{i+j}{2} \in \left[k + \frac{1}{2} - N\varepsilon, k + \frac{1}{2} + N\varepsilon\right].$$

This occurs with probability at most $\frac{2N\varepsilon}{N} = 2\varepsilon$.

2. Convergence in the i.i.d. setting. This follows from Hoeffding's inequality.

$\qquad\square$

# B  Proofs for Section 3

## B.1  Sandwiching covers

**Lemma 13** (Sandwich number for balls centered at $x$). *Let $(\mathcal{X}, \rho, \upsilon)$ be a separable metric space with a Borel probability measure. Fix $x \in \mathcal{X}$ and let $\mathcal{B}_x$ be the set of closed balls centered at $x$. Then, for any $\alpha \in [0,1]$, the $\alpha$-sandwich number $\mathcal{N}_{\mathcal{B}_x}(\alpha)$ of $\mathcal{B}_x$ is at most $4/\alpha$.*

*Proof.* Let $X \sim \upsilon$ and let $F(r)$ be the cumulative distribution function of $\rho(x, X)$. Let $M \in \mathbb{N}$ be any number greater than or equal to $1/\alpha$. For $m = 0, \ldots, M$, define:

$$r_m = \min\, \{r \geq 0 : F(r) \geq m/M\},$$

where $r_m$ exists because $F$ is upper semi-continuous and is possibly infinite. We claim that the following collection of open and closed balls forms an $\alpha$-sandwiching cover of $\mathcal{B}_x$,

$$\mathcal{C} = \bigcup_{m=0}^{M} \{B(x, r_m), \bar{B}(x, r_m)\},$$

from which the result follows by letting $M = \lceil \frac{1}{\alpha} \rceil$, since $4/\alpha \geq 2 \cdot \lceil \frac{1}{\alpha} + 1 \rceil \geq |\mathcal{C}|$.

We now choose $A_{\mathrm{in}}, A_{\mathrm{out}} \in \mathcal{C}$ satisfying the $\alpha$-sandwiching condition for any $\bar{B}(x, r) \in \mathcal{B}_x$. Let $m \in \{0, \ldots, M\}$ be the smallest number such that $F(r) \leq F(r_m)$, which implies:

$$\upsilon\big(\bar{B}(x, r_{m-1})\big) < \upsilon\big(\bar{B}(x, r)\big) \leq \upsilon\big(\bar{B}(x, r_m)\big).$$

There are two cases:

(a) If $r = r_m$, then we let $A_{\mathrm{in}} = A_{\mathrm{out}} = \bar{B}(x, r_m)$. The sandwiching condition evidently holds.

(b) If $r < r_m$, then we let $A_{\mathrm{in}} = \bar{B}(x, r_{m-1})$ and $A_{\mathrm{out}} = B(x, r_m)$. By construction of $r_m$,

$$\upsilon\big(\bar{B}(x, s)\big) < m/M, \qquad \forall s < r_m.$$

By the continuity of measure, we obtain $\upsilon\big(B(x, r_m)\big) \leq m/M$. By construction of $r_{m-1}$,

$$\upsilon\big(\bar{B}(x, r_{m-1})\big) \geq (m-1)/M.$$

It follows that $\upsilon\big(A_{\mathrm{out}} \setminus A_{\mathrm{in}}\big) \leq 1/M \leq \alpha$.

$\square$

## B.2  Concentration inequalities

**Lemma 10** (Concentration for label-oblivious queries). *Suppose that $(X_1, Y_1, \ldots, X_N, Y_N)$ is an adaptively sampled dataset and $Q$ is a label-oblivious query such that $|Q| \leq k$ almost surely. Then:*

$$\Pr\left(\big|\hat{\eta}_N(Q) - \bar{\eta}_N(Q)\big| \geq t\right) \leq 2k \exp\left(-2\,|Q|\,t^2\right).$$

*Proof.* Let $\tau_0 = 0$. Define $\tau_1, \tau_2, \ldots, \tau_k$ to be the sequence of stopping times:

$$\tau_i = \min\, \{t > \tau_{i-1} : t \in Q\},$$

where the stopping times are possibly infinite. Thus, $\tau_i$ is the $i$th index inserted into $Q$, and we have:

$$Q = \{\tau_i : \tau_i < \infty\}.$$

We define $Y_\infty = 0$, so that $Y_{\tau_i} = 0$ whenever $\tau_i = \infty$. Then, for any fixed $\ell \leq k$, the following forms a martingale difference sequence:

$$Y_{\tau_i} - \mathbb{E}\big[Y_{\tau_i} \,\big|\, X_{\tau_1}, Y_{\tau_2}, \ldots, X_{\tau_i}\big], \qquad i = 1, \ldots, \ell,$$

where we can let $X_\infty$ be defined as any deterministic constant.

Now, we may apply Azuma-Hoeffding's inequality, we obtain:

$$\Pr\left(\left|\sum_{i=1}^{\ell} Y_{\tau_i} - \sum_{i=1}^{\ell} \eta(X_{\tau_i})\right| \geq t\ell\right) \leq 2\exp\left(-2\ell t^2\right).$$

We can union bound over all $\ell = 1, \ldots, k$, and since $|Q| \leq k$ almost surely, the bound holds in particular for the random sample size $|Q|$,

$$\Pr\left(\left|\sum_{i=1}^{|Q|} Y_{\tau_i} - \sum_{i=1}^{|Q|} \eta(X_{\tau_i})\right| \geq t \cdot |Q|\right) \leq 2k\exp\left(-2|Q|t^2\right).$$

We obtain the result by a change of notation. Recall that when $|Q| > 0$,

$$\hat{\eta}_N(Q) = \frac{1}{|Q|} \sum_{i \in Q} Y_i \qquad \text{and} \qquad \bar{\eta}_N(Q) = \frac{1}{|Q|} \sum_{i \in Q} \eta(X_i),$$

and that when $|Q| = 0$, then $\hat{\eta}_N(Q) = \bar{\eta}_N(Q) = 0$. $\qquad\square$

**Lemma 14** (Uniform concentration for spatial queries). *Let* $(X_1, Y_1, \ldots, X_N, Y_N)$ *be an adaptively sampled dataset on* $(\mathcal{X}, \rho, \upsilon)$, *a metric measure space where* $\upsilon$ *is a Borel probability measure. Let* $\mathcal{A}$ *be a family of measurable sets and let* $\mathcal{Q}_{N,k}(\mathcal{A})$ *be a corresponding class of spatial queries. Suppose that* $\mathbb{X}$ *is* $\varepsilon(\delta)$-*uniformly dominated. For any* $p \in (0, 1)$, *with probability at least* $1 - p$,

$$\forall Q \in \mathcal{Q}_{N,k}(\mathcal{A}), \qquad \left|\hat{\eta}_N(Q) - \bar{\eta}_N(Q)\right| \leq \frac{2\ell}{|Q|} + \sqrt{\frac{1}{2(|Q| - \ell)} \log \frac{k \cdot \mathcal{N}_{\mathcal{A}}(\alpha)}{p}},$$

*whenever* $\alpha$ *satisfies* $\varepsilon(\alpha) = 1/N$ *and* $\ell \geq \max\{2\log\frac{2\mathcal{N}_{\mathcal{A}}(\alpha)}{p}, e^2\}$.

*Proof.* Let $\mathcal{C}$ be a minimal $\alpha$-sandwiching cover of $\mathcal{A}$, so that $|\mathcal{C}| \leq \mathcal{N}_{\mathcal{A}}(\alpha)$. By a union bound taken over queries $Q \in \mathcal{Q}_{N,k}(\mathcal{C})$, we obtain from Lemma 10 that with probability at least $1 - p/2$,

$$\forall Q \in \mathcal{Q}_{N,k}(\mathcal{C}), \qquad \left|\hat{\eta}_N(Q) - \bar{\eta}_N(Q)\right| \leq \sqrt{\frac{1}{2|Q|} \log \frac{k \cdot \mathcal{N}_{\mathcal{A}}(\alpha)}{p}}. \qquad (3)$$

To extend this large-deviation bound to the rest of $\mathcal{Q}_{N,k}(\mathcal{A})$, we shall use the fact that every $A \in \mathcal{A}$ is sandwiched between two elements of $\mathcal{C}$ that are $\alpha$-close $A_{\text{in}} \subset A \subset A_{\text{out}}$. We just need to ensure that the difference region $A_{\text{in}} \Delta A_{\text{out}}$ does not contain a very large number of points from $\mathbb{X}_{\leq N}$.

To do so, define the collection of difference regions between $\alpha$-close sets in $\mathcal{C}$:

$$\Delta_{\alpha}\mathcal{C} = \{A_1 \Delta A_2 : A_1, A_2 \in \mathcal{C} \text{ and } \upsilon(A_1 \Delta A_2) \leq \alpha\}.$$

There are at most $\mathcal{N}_{\mathcal{A}}(\alpha)^2$ such regions. We can now apply Lemma B.1 to show that when $\mathbb{X}$ is uniformly dominated, these regions rarely contains more than $\ell$ points. In particular, we take a union bound over all $\alpha$-close pairs of sets in $\mathcal{C}$, so that with probability at least $1 - p/2$,

$$\forall U \in \Delta_{\alpha}\mathcal{C}, \qquad \left|U \cap \mathbb{X}_{\leq N}\right| \leq \ell, \qquad (7)$$

whenever $\ell \geq \max\left\{2\log\frac{2\mathcal{N}_{\mathcal{A}}(\alpha)}{p}, e^2\right\}$ and $\varepsilon(\alpha) = 1/N$.

Given $A \in \mathcal{A}$, let $A_{\text{in}} \subset A \subset A_{\text{out}}$ satisfy the $\alpha$-sandwiching property. Let $Q_{\text{in}} = Q_{N,k}(A_{\text{in}})$. It follows that if both events (3) and (7) occur, then:

$$\sum_{i \in Q} Y_i - \eta(X_i) = \sum_{i \in Q_{\text{in}}} Y_i - \eta(X_i) + \sum_{i \in Q \setminus Q_{\text{in}}} Y_i - \eta(X_i) - \sum_{i \in Q_{\text{in}} \setminus Q} Y_i - \eta(X_i)$$

$$\leq |Q_{\text{in}}| \cdot \sqrt{\frac{1}{2|Q_{\text{in}}|} \log \frac{k \cdot \mathcal{N}_{\mathcal{A}}(\alpha)}{p}} + 2\ell,$$

where we use the fact that neither $Q \setminus Q_{\text{in}}$ nor $Q_{\text{in}} \setminus Q$ can contain be more than $\ell$ indices (note that $Q_{\text{in}} \setminus Q$ can be non-empty if $A$ reached capacity with some instances falling in $A \setminus A_{\text{in}}$). Dividing through by $|Q|$ yields the result, where we use $|Q_{\text{in}}| \leq |Q|$ and $1/|Q_{\text{in}}| \leq 1/(|Q| - \ell)$. $\qquad\square$

**Lemma B.1.** *Let $\mathbb{X}$ be $\varepsilon(\delta)$-uniformly dominated by $\upsilon$. Fix $N \in \mathbb{N}$ and let $A$ be measurable. Then:*

$$\Pr\left(\sum_{i=1}^{N} \mathbb{1}\{X_i \in A\} \geq \ell\right) \leq p,$$

*whenever $\ell \geq \max\left\{\log\frac{1}{p}, e^2 N\gamma\right\}$ and $\gamma = \varepsilon(\upsilon(A))$.*

*Proof.* The probability that $X_t \in A$ is bounded by $\alpha$ because $\mathbb{X}$ is uniformly dominated. It follows that for any fixed sequence of $\ell$ distinct times $1 \leq t_1 < t_2 < \cdots < t_\ell \leq N$, the probability that the event $X_{t_i} \in A$ occurs at each of these times is bounded by $\gamma^\ell$. Formally:

$$\Pr\left(\bigwedge_{i=1}^{\ell} X_{t_i} \in A\right) \leq \prod_{i=1}^{\ell} \Pr\left(X_{t_i} \in A \,\Big|\, \bigwedge_{j=1}^{i-1} X_{t_j} \in A\right)$$

$$\leq \prod_{i=1}^{\ell} \mathbb{E}\left[\Pr\left(X_{t_i} \in A \,|\, \mathbb{X}_{<t_i}\right) \,\Big|\, \bigwedge_{j=1}^{i-1} X_{t_j} \in A\right] \leq \gamma^\ell. \qquad (8)$$

We can bound the event that $\mathbb{X}_{\leq N}$ hits $A$ at least $\ell$ times by a union bound over all $\binom{N}{\ell}$ possible sets of time indexes $\{t_1, \ldots, t_\ell\}$ for which $X_{t_i} \in A$:

$$\Pr\left(\sum_{i=1}^{N} \mathbb{1}\{X_i \in A\} \geq \ell\right) \overset{(i)}{\leq} \sum_{1 \leq t_1 < \cdots < t_\ell \leq N} \Pr\left(\bigwedge_{i=1}^{k} X_{t_i} \in A\right)$$

$$\overset{(ii)}{\leq} \binom{N}{\ell}\gamma^\ell$$

$$\overset{(iii)}{\leq} \left(\frac{eN}{\ell}\right)^\ell \gamma^\ell$$

$$\overset{(iv)}{\leq} \frac{1}{e^\ell}$$

$$\overset{(v)}{\leq} p,$$

which follows from (i) the union bound, (ii) Equation 8, (iii) the standard bound $\binom{N}{\ell} \leq (eN/\ell)^\ell$, (iv) $\ell \geq e^2 N\gamma$, and (v) $\ell \geq \log\frac{1}{p}$. $\qquad \square$

### B.3 Concentration for $k$-nearest neighbor query

The following concentration for the $k$-nearest neighbor query for a random point $Z \sim \upsilon$ follows directly from Lemma 14, where $\mathcal{A}$ is the set $\mathcal{B}_Z$ of balls centered at $Z$ and a bound on its sandwich number comes from Lemma 13. A version with tie-breaking is given in Appendix E.

**Corollary 2** (Concentration for the $k$-nearest neighbor query). *Let $(X_1, Y_1, X_2, \ldots, X_N, Y_N)$ be adaptively sampled by an $\varepsilon(\delta)$-uniformly dominated process. Let $Z \sim \upsilon$ be independently sampled. Suppose that almost surely there are no distance ties to $Z$; let $X^{(1)}, \ldots, X^{(N)}$ sort the instances:*

$$\rho(Z, X^{(1)}) < \cdots < \rho(Z, X^{(N)}).$$

*Let $t, p \in (0, 1)$. Suppose that $k \geq \frac{100}{t^2}\left(1 + \log k + \log\frac{1}{\varepsilon^{-1}(1/N)} + \log\frac{1}{p}\right)$. Then:*

$$\Pr\left(\left|\frac{1}{k}\sum_{j=1}^{k} Y^{(j)} - \frac{1}{k}\sum_{j=1}^{k} \eta(X^{(j)})\right| \geq t\right) \leq p.$$

*Proof.* Since $Z$ is chosen independently from the data, we may apply the uniform convergence result Lemma 14 to $\mathcal{B}_Z$, the closed balls around $Z$. In particular, when $k$ is sufficiently large, then:

$$k > \frac{4\ell + \log k}{t^2} \quad \implies \quad \frac{2\ell}{k} + \sqrt{\frac{1}{2(k-\ell)}\log\frac{k \cdot \mathcal{N}_{\mathcal{B}_Z}(\alpha)}{p}} < t,$$

when $\varepsilon(\alpha) = 1/N$ and $\ell \geq \max\left\{2\log\frac{2\mathcal{N}_{\mathcal{B}_Z}(\alpha)}{p}, e^2\right\}$. By Lemma 14, with probability at least $1 - p$,

$$\left|\frac{1}{k}\sum_{j=1}^{k}Y^{(j)} - \frac{1}{k}\sum_{j=1}^{k}\eta(X^{(j)})\right| < t.$$

Then, Lemma 13 bounds $\mathcal{N}_{\mathcal{B}_Z}(\alpha) \leq 4/\alpha$. So indeed, $k$ is sufficiently large:

$$
\begin{aligned}
\frac{4\ell + \log k}{t^2} &\leq \frac{8}{t^2}\left(\log 2 + e^2 + \log k + \log\frac{4}{\alpha} + \log\frac{1}{p}\right) \\
&\leq \frac{8}{t^2}\left(10 + \log k + \log\frac{1}{\varepsilon^{-1}(1/N)} + \log\frac{1}{p}\right) < k,
\end{aligned}
$$

since $\log 2 + e^2 + \log 4 \leq 10$. $\qquad\square$

## B.4 Technical lemmas

**Lemma B.3** (Borel-Cantelli). *Let $(A_n)_n$ be a sequence of events such that $\sum\Pr(A_n) < \infty$. Then:*

$$\Pr\left(A_n \text{ occurs infinitely often}\right) = 0.$$

For reference, see Theorem 2.3.1 of Durrett (2019).

**Definition B.4** (Ergodic continuity). A stochastic process is *ergodically dominated* by $\upsilon$ if for any $\varepsilon > 0$, there exists $\delta > 0$ such that when a measurable set $A \subset \mathcal{X}$ satisfies $\upsilon(A) < \delta$, then:

$$\limsup_{N\to\infty}\frac{1}{N}\sum_{n=1}^{N}\mathbb{1}\{X_n \in A\} < \varepsilon \qquad \text{a.s.}$$

We say that $\mathbb{X}$ is *ergodically continuous* with respect to $\upsilon$ at rate $\varepsilon(\delta)$.

**Lemma B.5** (Lemma 5, Dasgupta and So (2024)). *Let $(X_n)_n$ be a process that is uniformly dominated by $\upsilon$ at rate $\varepsilon(\delta)$, and let $(\mathcal{F}_n)_n$ be its natural filtration. Let $(A_n)_n$ be a $(\mathcal{F}_n)_n$-predictable sequence where $\limsup_{n\to\infty}\upsilon(A_n) < \delta$. Then:*

$$\limsup_{N\to\infty}\frac{1}{N}\sum_{n=1}^{N}\mathbb{1}\{X_n \in A_n\} \leq \varepsilon(\delta) \qquad \text{a.s.}$$

**Lemma B.6** (Closeness of $k$-nearest neighbors). *Let $\mathbb{X}$ be any process in $\mathcal{X}$ and fix some $n \in \mathbb{N}$. Let $B = B(z, r)$ be any ball and let $3B$ denote the larger ball $B(z, 3r)$. Suppose that $X_n \in B$ and $|B \cap \mathbb{X}_{<n}| \geq k$. Then, the $k$-nearest neighbors of $X_n$ are contained in $3B$,*

$$X_n^{(1)}, \ldots, X_n^{(k)} \in 3B.$$

*Proof.* For all $x \in B$ and $x' \notin 3B$, we have:

$$\rho(x, X_n) \leq 2r \qquad \text{and} \qquad \rho(z, X_n) > 2r.$$

That is, all points in $B$ are closer to $X_n$ than points outside of $3B$. Since $B$ contains at least $k$ points, the $k$ nearest neighbors of $X_n$ must come from $3B$. $\qquad\square$

# C  Proofs for Section 4

**Theorem 15** (Consistency of the $k_n$-nearest neighbor rule)**.** *Let $(\mathcal{X}, \rho, \upsilon)$ be a separable metric space with a finite Borel measure. Let $\eta : \mathcal{X} \to [0,1]$ be continuous. Suppose that $\mathbb{X}$ is uniformly dominated at rate $\varepsilon(\delta)$ and that almost surely there are no distance ties. If $k_n$ satisfies (2), then the $k_n$-nearest neighbor rule is consistent almost surely.*

*Proof.* It suffices to show that the rate at which the $k_n$-nearest neighbor rule does not coincide with the Bayes optimal prediction goes to zero. When $\eta(X_n) = 1/2$, we say that both possible predictions in $\mathcal{Y} = \{0, 1\}$ are Bayes optimal. Then, the Bayes-inconsistent event is contained in:

$$\left\{ \hat{Y}_n \neq Y_n^* \right\} \subset \left\{ \left| \frac{1}{k_n} \sum_{k=1}^{k_n} Y_n^{(i)} - \eta(X_n) \right| \geq s \right\} \cup \left\{ 0 < \left| \eta(X_n) - \frac{1}{2} \right| \leq s \right\},$$

for any $s > 0$. By applying the triangle inequality to the statistical and geometric convergence results, Propositions 16 and 18, we obtain that:

$$\limsup_{N \to \infty} \frac{1}{N} \sum_{n=1}^{N} \mathbb{1} \left\{ \left| \frac{1}{k_n} \sum_{k=1}^{k_n} Y_n^{(i)} - \eta(X_n) \right| \geq s \right\} = 0 \qquad \text{a.s.}$$

Define $A_s$ to be the set:

$$A_s = \left\{ x \in \mathcal{X} : 0 < \left| \eta(x) - \frac{1}{2} \right| \leq s \right\}$$

Thus, the rate that $k_n$-nearest neighbor is inconsistent with the Bayes predictor is bounded by:

$$\limsup_{N \to \infty} \frac{1}{N} \sum_{n=1}^{N} \mathbb{1} \left\{ \hat{Y}_n \neq Y_n^* \right\} \leq \limsup_{N \to \infty} \frac{1}{N} \sum_{n=1}^{N} \mathbb{1} \left\{ X_n \in A_s \right\} \leq \varepsilon \left( \upsilon \left( A_s \right) \right),$$

where the second inequality holds by the uniform domination condition. The result follows by letting $s \downarrow 0$, which implies that $A_s \downarrow \varnothing$. By the continuity of measure, $\upsilon(A_s)$ also converges to zero, so by uniform domination, so does $\varepsilon(\upsilon(A_s))$. $\qquad \square$

# D  Proofs for Section 5

## D.1  Proof of Theorem 5

**Theorem 5** (Universal consistency in upper doubling spaces). *Let $(k_n)_n$ be a regular sequence, and $(\mathcal{X}, \rho, \upsilon)$ be an upper doubling space. Let $\mathbb{X}$ be uniformly dominated by $\upsilon$ and $\eta : \mathcal{X} \to [0, 1]$ be measurable. Then the $k_n$-nearest neighbor rule is online consistent with respect to $(\mathbb{X}, \eta)$.*

*Proof.* Fix $\gamma > 0$, and let $\mathbb{I}$ and $\eta_0$ be as described in Lemma 20. We show that the asymptotic mistake rate is almost surely bounded by $3\gamma$. This implies the result since $\gamma$ was arbitrary.

By construction, the subsequence $\mathbb{X}[1 - \mathbb{I}]$ asymptotically all but a $\gamma$-fraction of $\mathbb{X}$. And so, even if the learner makes a mistake at each indicated time $I_n = 1$, this would contribute at most an additive term of $\gamma$ in the overall mistake rate. In the remainder, we show that mistakes made during the rest of times, when $I_n = 0$, contribute at most another term of $2\gamma$.

As in Theorem 15, it suffices to show that the rate at which the $k_n$-nearest neighbor rule does not coincide with the Bayes optimal prediction goes to zero. On a non-indicated time $I_n = 0$, we have that $\eta(X_n) = \eta_0(X_n)$, and so for any $s > 0$, the Bayes-inconsistent event is contained in:

$$\left\{ \hat{Y}_n \neq Y_n^* \text{ and } I_n = 0 \right\}$$

$$\subset \quad \underbrace{\left\{ \left| \frac{1}{k_n} \sum_{k=1}^{k_n} Y_n^{(i)} - \eta_0(X_n) \right| \geq s \text{ and } I_n = 0 \right\}}_{(a)} \cup \underbrace{\left\{ 0 < \left| \eta_0(X_n) - \frac{1}{2} \right| \leq s \right\}}_{(b)}.$$

Let's bound the rate at which either (a) or (b) occurs. As $\eta_0$ is continuous, we may choose $s > 0$ such that by uniform domination, (b) contributes at most another $\gamma$ term to the mistake rate:

$$\limsup_{N \to \infty} \frac{1}{N} \sum_{n=1}^{N} \mathbb{1} \left\{ 0 < \left| \eta_0(X_n) - \frac{1}{2} \right| \leq s \right\} \leq \gamma \qquad \text{a.s.}$$

As for the event (a), we can first bound $\left| \frac{1}{k_n} \sum_{k=1}^{k_n} Y_n^{(i)} - \eta_0(X_n) \right|$ using triangle inequality by:

$$\underbrace{\left| \frac{1}{k_n} \sum_{k=1}^{k_n} Y_n^{(i)} - \frac{1}{k_n} \sum_{k=1}^{k_n} \eta(X_k) \right|}_{(i)} + \underbrace{\left| \frac{1}{k_n} \sum_{k=1}^{k_n} \eta(X_k) - \frac{1}{k_n} \sum_{k=1}^{k_n} \eta_0(X_k) \right|}_{(ii)} + \underbrace{\left| \frac{1}{k_n} \sum_{k=1}^{k_n} \eta_0(X_k) - \eta_0(X_n) \right|}_{(iii)},$$

where both (i) and (iii) eventually never exceed $s/3$ almost surely, resepctively by statistical convergence (Proposition 16) and geometric convergence (Proposition 18). Thus, to bound the rate that event (a) occurs, it suffices to bound the rate that (ii) exceeds $s/3$. This is bounded by the rate that an $s/3$-fraction of the $k_n$-nearest neighbors of $X_n$ come from the region $\{\eta \neq \eta_0\}$, and thus are indicated instances. To bound this, let $A_n$ indicate the event that $I_n = 0$ but at least an $s/3$-fraction of the $k_n$-nearest neighbors of $X_n$ are in $\mathbb{X}[\mathbb{I}]$. Then, it suffices to show:

$$\limsup_{N \to \infty} \frac{1}{N} \sum_{n=1}^{N} A_n = 0 \qquad \text{a.s.} \tag{9}$$

To do so, we bound the long-term influence of $\mathbb{X}\big[\mathbb{I}\big]$. Let $B_n$ be the indicator from Lemma 21:

$$B_n = \mathbb{1} \left( \exists X_m \in \left\{ X_n^{(1)}, \ldots, X_n^{(k_n)} \right\} : I_m J_{m,n} = 1 \right),$$

the event that some $k_n$-nearest neighbor of $X_n$ comes from $\mathbb{X}\big[\mathbb{I}\big]$ and is sampled by $\mathbb{J}$, which was defined in (6). By setting $\delta = \gamma$ and sending $\gamma \to 0$ as before, Lemma 21 shows that:

$$\limsup_{N \to \infty} \frac{1}{N} \sum_{n=1}^{N} B_n = 0 \qquad \text{a.s.} \tag{10}$$

To complete our proof, we relate $A_n$ to $B_n$, showing that a significant fraction of the $k_n$-nearest neighbors cannot come from $\mathbb{X}[\mathbb{I}]$, for otherwise they would have been sampled by $\mathbb{J}$. Observe that at time $n$, the conditional expectation of $B_n$ can be lower bounded in terms of $A_n$:

$$\mathbb{E}\big[B_n \,\big|\, \mathbb{X}_{<n}\big] = \Pr\left(\exists X_m \in \big\{X_n^{(1)}, \ldots, X_n^{(k_n)}\big\} : I_m J_{m,n} = 1\right)$$

$$\geq A_n \left(1 - \frac{1}{k_n}\right)^{k_n \cdot s/3}$$

$$\geq A_n \left(\frac{1}{4}\right)^{s/3},$$

where the last inequality holds for all $k_n \geq 2$. Now, averaging over time, we obtain:

$$\frac{1}{N} \sum_{n=1}^{N} \mathbb{E}\big[B_n \,\big|\, \mathbb{X}_{<n}\big] \geq \frac{1}{N} \left(\frac{1}{4}\right)^{s/3} \sum_{n=1}^{N} A_n.$$

By the martingale law of large numbers and Equation (10), as $N$ goes to infinity, the left-hand side must converge to zero almost surely. It follows that the right-hand side must as well. This implies Equation (9) and completes the proof. $\qquad\square$

### D.2 Proof of Lemma 20

**Lemma 20** (Reduction to Lipschitz setting). *Let $\upsilon$ be a probability measure on $\mathcal{X}$, and $\eta : \mathcal{X} \to [0,1]$ be measurable. Let $\mathbb{X}$ be uniformly dominated by $\upsilon$. For any $0 < \gamma < 1/2$, there exists $L > 0$, a continuous map $\eta_0 : \mathcal{X} \to [0,1]$, and an indicator process $\mathbb{I}$ adapted to $\mathbb{X}$ such that $\mathbb{I}$ is asymptotically $\gamma$-rate-limited, the subsequence $\mathbb{X}[1 - \mathbb{I}]$ is L-dominated by $\upsilon$, and $\eta\big(\mathbb{X}[1 - \mathbb{I}]\big) = \eta_0\big(\mathbb{X}[1 - \mathbb{I}]\big)$.*

*Proof.* Let $\mu_n$ denote the conditional measure of $X_n$ on $\mathcal{X}$, so that for all measurable $A \subset \mathcal{X}$:

$$\mu_n(A) = \Pr\big(X_n \in A \,|\, \mathbb{X}_{<n}, \mathbb{Y}_{<n}\big).$$

Because $\mathbb{X}$ is uniformly dominated by $\upsilon$, the measure $\mu_n$ itself is absolutely continuous with respect to $\upsilon$. Thus, the Radon-Nikodym theorem implies that it has a density function $f_n : \mathcal{X} \to [0, \infty)$,

$$\mu_n(A) = \int_A f_n(x) \, \upsilon(dx).$$

Let $L = \frac{2}{\varepsilon^{-1}(\gamma/2)}$ and let $A_n = \big\{x : f_n(x) \geq \frac{L}{2}\big\}$. By design, $X_n$ is unlikely to be drawn from $A_n$,

$$\mu_n(A_n) \overset{(i)}{\leq} \varepsilon\big(\upsilon(A_n)\big) \overset{(ii)}{=} \varepsilon\left(\Pr_{X \sim \upsilon}\left(f_n(X) \geq \frac{L}{2}\right)\right) \overset{(iii)}{\leq} \varepsilon\left(\frac{2}{L} \cdot \mathop{\mathbb{E}}_{X \sim \upsilon}[f_n(X)]\right) \overset{(iv)}{\leq} \frac{\gamma}{2},$$

where (i) applies the uniform domination property of $\mu_n$, (ii) rewrites $\upsilon(A_n)$ as a probability, (iii) applies Markov's inequality, and (iv) follows from our choice of $L$ and from the fact that $f_n$ is a density function, so that $\mathbb{E}_{X \sim \upsilon}[f_n(X)] = 1$.

We now also construct the continuous map $\eta_0$. By Lusin's theorem, there is an open subset $B \subseteq \mathcal{X}$ with $\upsilon(B) < \varepsilon^{-1}\left(\frac{\gamma}{2}\right)$ such that the restriction of $\eta$ to $\mathcal{X} \setminus B$ is continuous. Since $\mathcal{X}$ is a metric space, this restriction can be continuously extended to all of $\mathcal{X}$ using the Tietze extension theorem. Let $\eta_0$ be any such extension. Then, $X_n$ is also unlikely to be drawn from $B$,

$$\mu_n(B) \leq \varepsilon\big(\upsilon(B)\big) \leq \frac{\gamma}{2}.$$

Finally, we define the indicator process $\mathbb{I} = (I_n)_n$, where:

$$I_n = \mathbb{1}\big\{X_n \in A_n \cup B\big\}.$$

Since $\mu_n(A_n \cup B) \leq \gamma$, the martingale law of large numbers implies that $\mathbb{I}$ is $\gamma$-rate limited:

$$\limsup_{N \to \infty} \frac{1}{N} \sum_{n=1}^{N} I_n \leq \gamma \qquad \text{a.s.},$$

proving that condition (1) holds. Condition (3) also follows by construction, since $\eta_0$ is equal to $\eta$ on $\mathcal{X} \setminus B$. Indeed, when $I_n = 1$, then $X_n \notin B$. Condition (2) remains.

For this last condition, for any $n$ where $I_n = 0$, the distribution from which $X_n$ is selected is conditioned on the fact that $X_n \notin A_n \cup B$. Since $A_n \cup B$ has $\upsilon$-measure at most $\gamma$, we see that for any measurable set $E$, the probability that $X_n \in E$ is at most:

$$
\begin{aligned}
\Pr\left(X_n \in E \mid \mathbb{X}_{<n}, \mathbb{Y}_{<n}, I_n = 0\right) &= \frac{\mu_n\left(E \setminus (A_n \cup B)\right)}{1 - \mu_n\left(A_n \cup B\right)} \\
&\leq \frac{1}{1-\gamma} \int_E f_n(x) \cdot \mathbb{1}\{x \notin (A_n \cup B)\}\, \upsilon(dx) \\
&\leq \frac{1}{1-\gamma} \int_E f_n(x) \cdot \mathbb{1}\{f_n(x) \leq L/2\}\, \upsilon(dx) \\
&\leq \frac{1}{1-\gamma} \int_E \frac{L}{2}\, \upsilon(dx) \\
&= \frac{L}{2(1-\gamma)} \cdot \upsilon(E) \leq L\upsilon(E).
\end{aligned}
$$

Here, we exploited the fact outside of the set $A_n$, the density $f_n$ is at most $L/2$, and that $\gamma < 1/2$. $\quad\square$

## D.3 Proof of Lemma 21

**Lemma 21** (Long-term influence bound). *Let $(\mathcal{X}, \rho, \upsilon)$ be an upper doubling space and $(k_n)_n$ a regular sequence. There exist $c_1, c_2 > 0$ so that the following holds. Let $\mathbb{X}$ be uniformly dominated at rate $\varepsilon(\delta)$ and $\mathbb{I}$ be an indicator process adapted to $\mathbb{X}$ asymptotically rate-limited by $\gamma > 0$, and $\mathbb{J}$ be given by (6). For any $\delta > 0$, the rate that an $\mathbb{I}$-indicated $k_n$-nearest neighbor is sampled by $\mathbb{J}$ is:*

$$
\limsup_{N \to \infty} \frac{1}{N} \sum_{n=1}^{N} \mathbb{1}\left(\exists X_m \in \{X_n^{(1)}, \dots, X_n^{(k_n)}\} : I_m J_{m,n} = 1\right) < \gamma\left(c_1 + c_2 \log \frac{1}{\delta}\right) + \varepsilon(\delta) \quad \text{a.s.}
$$

The proof of this result builds on the technique developed by Dasgupta and So (2024), specifically for their Theorem 17. In Sections D.4 and D.5, we review the key definitions from their work along with some modifications to extend to $k_n$-nearest neighbors (rather than 1-nearest neighbors).

We begin with a quick technical lemma that lets us bound how many points are likely to be selected for which both $I$ and $J$ are 1.

**Lemma D.1.** *Let $\mathbb{I}$ and $\mathbb{J}$ be as defined in Lemma 21. Then:*

$$
\limsup_{N \to \infty} \frac{k_N}{N} \sum_{n=1}^{N} I_n J_{n,N} \leq \gamma.
$$

*Proof.* It is clear that $\limsup_{N \to \infty} \frac{1}{N} \sum_{n=1}^{N} I_n \leq \gamma$ almost surely. Thus, for any $\alpha > 0$, there almost surely exists $N_\alpha$ such that for all $N \geq N_\alpha$, $\frac{1}{N} \sum I_n \leq \gamma + \alpha$.

Pick any $N$ for which this occurs. Observe that the values of $J_{n,N}$ are i.i.d Bernouli variables each with expected value $k_N$. Thus applying Hoeffding's lemma over all of them, we see that with probability at least $1 - \exp\left(-\frac{N(\gamma+\alpha)}{K_N}\right)$, our desired sum is at most $(\gamma + \alpha)(1 + \alpha)$.

Since $(k_n)_n$ is a regular sequence, $\frac{N}{K_N}$ grows strictly faster than $\log n$, which implies that $\sum_{N=1}^{\infty} \exp\left(-\frac{N(\gamma+\alpha)}{K_N}\right)$ is finite. Applying the Borel-Cantelli lemma implies that our deisred sum is at most $(\gamma + \alpha)(1 + \alpha)$ almost surely completing the proof. $\quad\square$

*Proof of Lemma 21.* Without loss of generality, $\mathcal{X}$ has diameter 1. For $1 \leq n \leq N$, let $\kappa(n)$ be the number of instances $1 \leq t \leq n$ for which $I_t J_{t,N} = 1$, and let $\tau_k$ to be the time of the $k$th time for which this occurs.

Next, we construct a chain of sequentially-constructed cover trees (Definition D.6) *exactly* as done in the proof of Theorem 17 in Dasgupta and So (2024). That is, let $(\mathcal{C}_k)_k$ be a chain of sequentially-constructed cover trees associated to the sequence $X_{\tau_1}, X_{\tau_2}, \ldots$. Let $L_k$ denote the insertion rank of $X_{\tau_k}$. For $n \geq \tau_k$, define

$$T_{k,n} = L_k + 1 + \left\lceil \frac{1}{d} \lg \frac{c}{\delta} \right\rceil + G_{k,\kappa(n)}, \tag{11}$$

where $(c, d)$ are parameters associated to the upper doubling condition (Definition 3), and $G_{k,\kappa(n)}$ is the number of generations of children $X_{\tau_k}$ has by time $n$.

Next, construct a set of very small balls that are centered at the points in our cover tree as follows. Define

$$\mathcal{T}_n = \bigcup_{k=1}^{\kappa(n)} \text{cone}(X_{\tau_k}; T_{k,n} + 1) \qquad \text{and} \qquad A_n = \bigcup_{B \in \mathcal{T}_n} 2B,$$

where the cone is defined as in Definition D.5.

We now relate the number of times in which $X_n$ has a nearest neighbor with $I_t J_{t,N} = 1$ to events related to this cover tree. Since $\upsilon$ is an upper doubling measure, we immediately have that the sequence $X_1, X_2, \ldots$ are almost surely distinct points. Thus, applying Lemma D.8, with the combined indicator process $I_n J_{n,N}$, we have

$$\sum_{n=1}^N \mathbb{1}\left( \exists X_t \in \left\{ X_n^{(1)}, \ldots, X_n^{(k_n)} \right\} \right) \leq \sum_{n=1}^N \sum_{B_r \in \mathcal{C}_{\kappa(n-1)}} \mathbb{1}\left( E_n^{2B_r, r/2} \right)$$

$$= \sum_{n=1}^N \sum_{B_r \in \mathcal{C}_{\kappa(n-1)} \setminus \mathcal{T}_{n-1}} \mathbb{1}\left( E_n^{2B_r, r/2} \right) + \sum_{n=1}^N \mathbb{1}\left( X_n \in A_{n-1} \right).$$

We now bound each term separately. For the first term, we apply Lemma D.4 to see that

$$\sum_{n=1}^N \sum_{B_r \in \mathcal{C}_{\kappa(n-1)} \setminus \mathcal{T}_{n-1}} \mathbb{1}\left( E_n^{2B_r, r/2} \right) \leq \sum_{B_r \in \mathcal{C}_{\kappa(N-1)} \setminus \mathcal{T}_{N-1}} \sum_{n=1}^N \mathbb{1}\left( E_n^{2B_r, r/2} \right) \leq 2^{2d+1} k_N |\mathcal{C}_{\kappa(N)} \setminus \mathcal{T}_N|.$$

Bounding the total number of balls within this set follows *identically* to the proof of Theorem 17 in Dasgupta and So (2024): it suffices to count the number of balls for each instance $X_{\tau_k}$ that are not in its $(T_{k,N} + 1)$-tail:

$$|\mathcal{C}_{\kappa(N)} \setminus \mathcal{T}_N| \leq \sum_{k=1}^{\kappa(N)} (T_{k,N} + 1) - L_k$$

$$\leq \kappa(N) \cdot \left( 2 + \left\lceil \frac{1}{d} \lg \frac{c}{\delta} \right\rceil \right) + \sum_{k=1}^{\kappa(N)} G_{k,\kappa(N)}$$

$$\leq \kappa(N) \cdot \left( 2 + \left\lceil \frac{1}{d} \lg \frac{c}{\delta} \right\rceil \right) + \kappa(N).$$

To bound the second term, Lemma D.10 implies that $A_n$ has mass at most $\upsilon(A_n) \leq \delta$ for all $n$. Thus the probability that $X_n \in A_{n-1}$ is at most $\varepsilon(\delta)$, and the expected number of total occurrences is $N(\varepsilon(\delta))$.

Combining our bounds, fix any $\alpha > 0$. Lemma 5 from Dasgupta and So (2024) implies that for $N$ sufficiently large, $\sum_{n=1}^N \mathbb{1}(X_n \in A_{n-1}) < N(\varepsilon(\delta) + \alpha)$ with probability 1. Furthermore, by applying Lemma D.1, we see that for $N$ sufficiently large, $\kappa(N) < \frac{N}{k_N}(\gamma + \alpha)$ also occurs with probability 1. Combining these, we see that

$$\frac{1}{N} \sum_{n=1}^N \mathbb{1}\left( \exists X_t \in \left\{ X_n^{(1)}, \ldots, X_n^{(k_n)} \right\} \right)$$

$$\leq 2^{2d+1} k_N \left( \frac{1}{k_N}(\gamma + \alpha) \right) \left( 3 + \left\lceil \frac{1}{d} \lg \frac{c}{\delta} \right\rceil \right) + (\varepsilon(\delta) + \alpha)$$

$$\leq (\gamma + \alpha) 2^{2d+1} \left( 3 + \left\lceil \frac{1}{d} \lg \frac{c}{\delta} \right\rceil \right) + \varepsilon(\delta) + \alpha.$$

Since $\alpha > 0$ was arbitrary, it follows that almost surely,

$$\limsup_{N \to \infty} \frac{1}{N} \sum_{n=1}^{N} \mathbb{1}\left(\exists X_t \in \left\{X_n^{(1)}, \ldots, X_n^{(k_n)}\right\}\right) \leq \gamma 2^{2d+1} \left(3 + \left\lceil \frac{1}{d} \lg \frac{c}{\delta} \right\rceil\right) + \varepsilon(\delta).$$

$\square$

## D.4 Packing Bounds

Here we include several definitions that are taken directly from Section 6.1 of Dasgupta and So (2024). In some cases, we include slight modifications that will prove useful for generalizing their results from 1-nearest neighbor to $k_n$-nearest neighbors.

**Definition D.2** (Packing number, Definition 18 of Dasgupta and So (2024)). Let $r > 0$. A set $Z \subset \mathcal{X}$ is an *r-packing* if all of its points are bounded away from each other by a distance $r$,

$$\inf_{z,z' \in Z} \rho(z, z') \geq r.$$

The *r-packing number* $\mathcal{P}_r(U)$ of $U$ is the maximum possible size of an $r$-packing $Z$ contained in $U$.

Similarly to Dasgupta and So (2024), we will use packing numbers to bound the number of times a point has a large $k_n$-nearest neighbor radius (its furthest neighbor is quite far from it). To do so, we adapt their notion of an "$r$-separated event" (Definition 19 in Dasgupta and So (2024)) to apply to $k_n$-nearest neighbors. An $r$-separated event is visualized in Figure 2.

**Definition D.3** ($r$-separated event). Let $\mathbb{X}$ be a process, $r > 0$, and $(k_n)_n$ be a sequence. The *r-separated event* at time $n$ is the event $E_n^r$ that $X_n$ has distance at least $r$ from its $k_n$th nearest neighbor. That is,

$$E_n^r := \left\{\rho(X_n, X_n^{(k_n)}) \geq r\right\}.$$

Given a subset $U$, the $(U, r)$-*separated events* are the events $E_n^{U,r} := E_n^r \cap \left\{X_n \in U\right\}$.

Next, similar in spirit to Lemma 20 of Dasgupta and So (2024), we bound how frequently $r$-separated events can occur with respect to the packing number.

**Lemma D.4** (Packing bound). *Let $(\mathcal{X}, \rho)$ be a metric space, $U \subset \mathcal{X}$ be a subset, $r > 0$, and $(k_n)_n$ be regular sequence (Definition 4). For any process $\mathbb{X}$, the number of $(U, r)$-separated events that occur before time $N$ is bounded by the $r$-packing number of $U$,*

$$\sum_{n=1}^{N} \mathbb{1}\left\{E_n^{U,r} \text{ occurs}\right\} \leq 2k_N \mathcal{P}_r(U).$$

*Proof.* Let $n_1, \ldots, n_t$ index all instances where the $(U, r)$-separated event $E_{n_i}^{U,r}$ occurs. Construct a graph with vertices $n_1, \ldots, n_t$ such that $(n_i, n_j)$ is a directed edge if and only if $n_i \leq n_j$ and $X_{n_i}$ was one of the $(k_{n_j} - 1)$-nearest neighbors of $X_{n_j}$. See also Figure 2. We make two claims:

1. Any independent set in this graph must form an $r$-packing of $U$.

2. The graph has an independent set of size $t/2k_N$.

Together, these two claims imply the result, since it shows that $t/2k_N \leq \mathcal{P}_r(U)$, recalling that $t$ is the total number of $(U, r)$-separated events that occur by time $N$. We now prove the claims:

- Claim 1. Let $n_i \leq n_j$. By construction, when $(n_i, n_j)$ is not an edge, the points $X_{n_i}$ and $X_{n_j}$ must be $r$-separated; this is because $X_{n_i}$ can be no closer to $X_{n_j}$ than the $k_{n_j}$th nearest neighbor of $X_{n_j}$. Thus, any independent set in the graph must form an $r$-packing of $U$.

- Claim 2. The graph has a total of at most $\sum_{i=1}^{t}(k_{n_i} - 1) \leq t(k_N - 1)$ edges ($k_N$ is larger than $k_{n_i}$ by the regularity of $(k_n)_n$). It follows that the average degree is at most $2(k_N - 1)$. Applying Turan's theorem, the graph must have an independent set of size at least:

$$\frac{t}{2k_N - 1} \geq \frac{t}{2k_N}.$$

$\square$

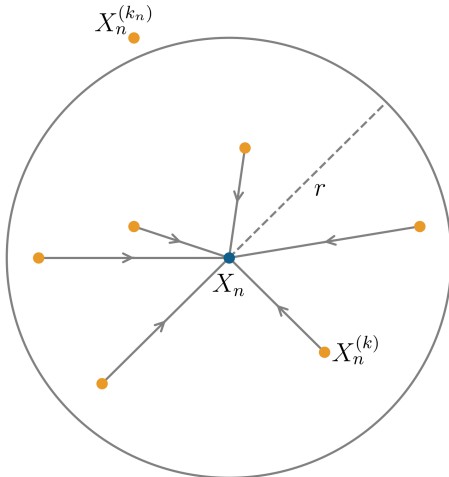

Figure 2: We say that an $r$-separated event occurs when the $k_n$th nearest neighbor of $X_n$ is $r$-separated from $X_n$. Thus, the ball $B(X_n, r)$ contains at most $k_n - 1$ neighbors.

## D.5 Properties of cover trees

Next, we review *cover trees*, which were introduced by (Beygelzimer et al., 2006) and a book-keeping device applied by Dasgupta and So (2024). We begin with several definitions and lemmas taken directly from Dasgupta and So (2024). We refer the interested reader to view their work for the intuition behind these ideas: for our purposes we will simply state them for completeness.

**Definition D.5** (Dyadic cone: Definition 21 from Dasgupta and So (2024)). Let $(\mathcal{X}, \rho)$ have unit diameter and let $x \in \mathcal{X}$. A *dyadic cone* centered at $x$ of rank $L \in \mathbb{Z}_{\geq 0}$ is the discrete collection of balls:

$$\mathrm{cone}(x; L) = \{B(x, 2^{-\ell}) : \ell \geq L \text{ and } \ell \in \mathbb{Z}_{\geq 0}\}.$$

When $L' \geq L$, we also refer to $\mathrm{cone}(x; L')$ within $\mathrm{cone}(x; L)$ as its *rank-$L'$ tail*.

**Definition D.6** (Sequentially-constructed cover trees: Definition 22 from Dasgupta and So (2024)). Let $(\mathcal{X}, \rho)$ have unit diameter and $\mathbb{A} = (a_k)_k$ be a dataset in $\mathcal{X}$ without duplicates. The *cover trees* $(\mathcal{C}_k)_k$ with *insertion ranks* $(L_k)_k$ are defined:

$$\mathcal{C}_1 = \mathrm{cone}(a_1; L_1), \qquad L_1 = 0,$$

$$\mathcal{C}_k = \mathcal{C}_{k-1} \cup \mathrm{cone}(a_k; L_k), \qquad L_k = \min\left\{\ell \in \mathbb{Z}_{\geq 0} : \text{no ball of radius } 2^{-\ell} \text{ in } \mathcal{C}_{k-1} \text{ contains } x\right\}.$$

We say that $a_k$ was *inserted* into the cover tree at the $L_k$th rank.

**Definition D.7** (Cover-Tree Neighbor Map: Equation 7 from Dasgupta and So (2024)). We define a *cover-tree neighbor* map $\mathfrak{c}_k : \mathcal{X} \setminus \mathbb{A}_{\leq k} \to \mathcal{C}_k$ as any one satisfying:

$$\mathfrak{c}_k(x) = B(a, r) \qquad \Longrightarrow \qquad x \in B(a, 2r) \quad \text{and} \quad r/2 \leq \rho(x, \mathbb{A}_{\leq k}) < r. \tag{12}$$

Dasgupta and So (2024) show that such maps always exist with the following lemma.

**Lemma D.8** (Lemma 23 from Dasgupta and So (2024)). *The cover tree $\mathcal{C}_k$ for $\mathbb{A}_{\leq k}$ has a cover-tree neighbor map $\mathfrak{c}_k$.*

**Definition D.9** (Tree structure: Definition 24 from Dasgupta and So (2024)). For the above sequence of cover trees and insertion ranks, we let $a_1$ be the *root* of $\mathbb{A}$. For all $k > 1$, there is a ball $B(a_j, 2^{-L_k+1}) \in \mathcal{C}_{k-1}$ containing $a_k$. We assign such an $a_j$ to be the *parent* of $a_k$, and we say that $a_k$ is its *child*. A set of instances inserted at the same rank to the same parent is called a *generation* of children. The number of generations of children $a_k$ has at time $n$ defines the upper triangular array $(G_{k,n})_{k \leq n}$:

$$G_{k,n} = \left|\left\{L_{k'} : a_k \text{ is the parent of } a_{k'} \text{ where } k' \leq n\right\}\right|.$$

**Lemma D.10** (Cover tree $\delta$-tail: Lemma E.1 from Dasgupta and So (2024)). *Let $(\mathcal{X}, \rho, \upsilon)$ be a upper doubling metric space with unit diameter. Let $(\mathcal{C}_k)_k$ be a chain of cover trees for the sequence*

$\mathbb{A} = (a_k)_k$ and let $(L_k)_k$ be its sequence of insertion ranks. Define the array of tail ranks $(T_{k,n})_{k \le n}$ and tail sets $(A_n)_n$,

$$T_{k,n} = L_k + 1 + \left\lceil \frac{1}{d} \lg \frac{c}{\delta} \right\rceil + G_{k,n} \qquad \text{and} \qquad A_n = \bigcup_{k=1}^{n} B\left(a_k, 2^{-T_{k,n}}\right),$$

*where $d$ is the doubling dimension and $c$ is the upper doubling constant in Definition 3. For all $n$, the mass of the tail is bounded $\upsilon(A_n) < \delta$.*

For the final lemma in this section, we will modify Lemma 25 from Dasgupta and So (2024) in order to apply for $k_n$-nearest neighbors (rather than only 1-nearest neighbor).

**Lemma D.11** (Cover tree decomposition). *Let $(\mathcal{X}, \rho)$ have unit diameter, let $\mathbb{X}$ be a process in $\mathcal{X}$, and let $\mathbb{I}$ be an indicator process. For any $n$, let $\mathcal{C}$ be a cover tree for $\mathbb{X}[\mathbb{I}_{<n}]$ with a cover-tree neighbor map $\mathfrak{c}$. Assume that $X_n \notin \mathbb{X}[\mathbb{I}_{<n}]$ is not equal to one of the indicated instances. Then:*

$$\left\{ \exists X_t \in \left\{ X_n^{(1)}, \dots, X_n^{(k_n)} \right\} \cap \mathbb{X}[\mathbb{I}_{<n}]] \right\} \subset \bigcup_{B(a,r) \in \mathcal{C}} \left\{ E_n^{r/2} = 1 \text{ and } \mathfrak{c}(X_n) = B(a,r) \right\},$$

*where $E_n^r$ (Definition D.3) denotes the event that $X_n$ has distance at least $r$ from its $k_n$-th nearest neighbor in $\mathbb{X}_{<n}$. In particular, the event within the union indexed by $B = B(a,r) \in \mathcal{C}$ is contained in $E_n^{2B,r/2}$.*

*Proof.* Let $\mathfrak{c}(X_n) = B(a,r)$ for some $r > 0$. Suppose that $X_n$ indeed has some nearest neighbor that is located within the indicated instances. By the definition of a cover tree neighbor, there must exist some $X_t \in \mathbb{X}[\mathbb{I}_{<n}]$ such that $\rho(X_n, X_t) \ge \frac{r}{2}$. This implies that the $k_n$-nearest neighbor distance of $X_n$ is at least $r/2$, which implies $E_n^{r/2}$ does indeed occur. Finally, the fact that $X_n \in 2B(a,r)$ is immediate from the definition of a cover tree neighbor. $\square$

# E Tie-breaking details

In this section, we formalize the tie-breaking strategy and show that the techniques introduced earlier also apply. Let $(X_1, Y_1, \ldots, X_N, Y_N)$ be an adaptively sampled dataset. For a new query point $x \in \mathcal{X}$, we sort the dataset $X^{(1)}, \ldots, X^{(N)}$ by the lexicographic ordering on the pair:

$$\big( \rho(x, X_i), i \big).$$

Thus, the ordering is foremost determined by the distance of an instance to $x$, so that $\rho(x, X^{(n)})$ is monotonically increasing with $n$. And if there are distance ties with $\rho(x, X^{(n)}) = \rho(x, X^{(n+1)})$, then the instance that entered the dataset earlier takes precedence. The $k_n$-*nearest neighbor query* consists of taking the first $k_n$ indices under this ordering.

It turns out that we can describe the $k_n$-nearest neighbor query as a relatively simple composition of spatial queries (Definition 8). In addition to the ball queries (Definition 11) defined earlier, we need the *shell queries*, which are spatial queries over sets of the form $S_r = \{z \in \mathcal{X} : \rho(x, z) = r\}$. Then, the $k_n$-nearest neighbor query (with tie-breaking) can be given as follows:

**Definition E.1** ($k_n$-nearest neighbor query)**.** Fix $x \in \mathcal{X}$ and $k \in [N]$. Let $(X_1, Y_1, \ldots, X_N, Y_N)$ be an adaptively sampled dataset. Then:

- Let $\mathcal{B}_x^\circ = \{B_r : r > 0\}$ be the set of open balls centered at $x$, where $B_r = B(x, r)$.

- Let $\mathcal{S}_x = \{S_r : r \geq 0\}$ be the set of shells centered at $x$, where $S_r = \bar{B}(x, r) \setminus B(x, r)$.

- The $k$-*nearest neighbor query* at $x$ is the adaptive query:

$$Q_{N,k}(x) = Q_{N,k}(B_r) \cup Q_{N,\ell}(S_r) \qquad \text{where} \quad r = \arg\max_{s > 0} \left\{ \big| Q_{N,k}(\bar{B}(x, s)) \big| \geq k \right\},$$

and where $\ell = k - |Q_{N,k}(B_r)|$. Thus, $Q_{N,k}(x)$ selects for exactly $k$ points by finding the smallest closed ball containing at least $k$ points, selecting for all points in its interior, and filling the remaining quota from the boundary, taken in order of arrival.

## E.1 Sandwich numbers and concentration

Notice that any uniform concentration for spatial queries over closed balls directly carries over to spatial queries over open balls: for every open ball $B_r \in \mathcal{B}_x^\circ$ and dataset $(X_1, Y_1, \ldots, X_N, Y_N)$, as long as $s < r$ is sufficiently close to $r$, then the queries coincide $Q_{N,k}(B_r) = Q_{N,k}(\bar{B}_s)$.

We need to provide uniform concentration of the class of spherical queries $\mathcal{Q}_{N,k}(\mathcal{S}_x)$. A very similar proof to Lemma 13 yields the following bound on the sandwich number of $\mathcal{S}_x$:

**Lemma E.2** (Sandwich number for spheres centered at $x$)**.** *Let $(\mathcal{X}, \rho, \upsilon)$ be a separable metric space with a Borel probability measure. Fix $x \in \mathcal{X}$ and let $\mathcal{S}_x$ be the set of shells centered at $x$. Then, for any $\alpha \in [0, 1]$, the $\alpha$-sandwich number $\mathcal{N}_{\mathcal{B}_x}(\alpha)$ of $\mathcal{S}_x$ is at most $4/\alpha$.*

*Proof.* Let $X \sim \upsilon$ and let $F(r)$ be the cumulative distribution function of $\rho(x, X)$. Let $M \in \mathbb{N}$ be any number greater than or equal to $1/\alpha$. For $m = 0, \ldots, M$, define:

$$r_m = \min \left\{ r \geq 0 : F(r) \geq m/M \right\},$$

where $r_m$ exists because $F$ is upper semi-continuous and is possibly infinite. We claim that the following collection of rings and shells forms an $\alpha$-sandwiching cover of $\mathcal{S}_x$,

$$\mathcal{C} = \bigcup_{m=0}^{M} \left\{ B(x, r_m) \setminus \bar{B}(x, r_{m-1}), S(x, r_m) \right\},$$

from which the result follows by letting $M = \lceil \frac{1}{\alpha} \rceil$, since $4/\alpha \geq 2 \cdot \lceil \frac{1}{\alpha} + 1 \rceil \geq |\mathcal{C}|$.

We now choose $A_{\text{in}}, A_{\text{out}} \in \mathcal{C}$ satisfying the $\alpha$-sandwiching condition for any $S(x, r) \in \mathcal{S}_x$. Let $m \in \{0, \ldots, M\}$ be the smallest number such that $F(r) \leq F(r_m)$. There are two cases:

(a) If $r = r_m$, then we let $A_{\text{in}} = A_{\text{out}} = S(x, r_m)$. The sandwiching condition evidently holds.

(b) If $r < r_m$, then we let $A_{\text{in}} = \varnothing$ and $A_{\text{out}} = B(x, r_m) \setminus \bar{B}(x, r_{m-1})$. By construction of $r_m$,

$$v\big(\bar{B}(x, s)\big) < m/M, \qquad \forall s < r_m.$$

By the continuity of measure, we obtain $v\big(B(x, r_m)\big) \le m/M$. By construction of $r_{m-1}$,

$$v\big(\bar{B}(x, r_{m-1})\big) \ge (m-1)/M.$$

It follows that $v\big(A_{\text{out}}\big) \le 1/M \le \alpha$.

$\square$

**Lemma E.3** (Concentration for the $k$-nearest neighbor query). *Let $(X_1, Y_2, \ldots, X_N, Y_N)$ be adaptively sampled by an $\varepsilon(\delta)$-uniformly dominated process. Let $Z \sim v$ be independently sampled. Let $X^{(1)}, \ldots, X^{(N)}$ sort the instances by ascending lexicographic ordering on the pair:*

$$\big(\rho(Z, X_i), n\big).$$

*Let $t, p \in (0, 1)$. Suppose that $k \ge \frac{500}{t^2}\left(1 + \log k + \log \frac{1}{\varepsilon^{-1}(1/N)} + \log \frac{1}{p}\right)$. Then:*

$$\Pr\left(\left|\frac{1}{k}\sum_{j=1}^{k} Y^{(j)} - \frac{1}{k}\sum_{j=1}^{k} \eta(X^{(j)})\right| \ge t\right) \le p.$$

*Proof.* Since $Z$ is chosen independently from the data and let $\mathcal{A} = \mathcal{B}_Z^\circ \cup \mathcal{S}_Z$ consist of the open balls and shells around $Z$. Thus, as shown in Definition E.1, the $k$-nearest neighbor query centered at $Z$ is the union of two disjoint queries:

$$Q_1, Q_2 \in \bigcup_{m \in [k]} \mathcal{Q}_{N,m}(\mathcal{A}).$$

The result follows by proving uniform convergence over queries that are the disjoint union of two such queries. As $Z$ is independently chosen from the data, we may apply the uniform convergence result Lemma 14. In particular, by a union bound, with probability at least $1 - p$,

$$\forall Q \in \bigcup_{m \in [k]} \mathcal{Q}_{N,m}(\mathcal{A}), \qquad \left|\hat{\eta}_N(Q) - \bar{\eta}_N(Q)\right| \le \frac{2\ell}{|Q|} + \sqrt{\frac{1}{2(|Q| - \ell)}\log\frac{k^2 \cdot \mathcal{N}_{\mathcal{A}}(\alpha)}{p}},$$

where we let $\alpha = \varepsilon^{-1}(1/N)$ and $\ell = 2\log\frac{2\mathcal{N}_{\mathcal{A}}(\alpha)}{p} + e^2$.

Let $Q_1$ and $Q_2$ be two disjoint queries satisfying the above inequality. Let $Q = Q_1 \cup Q_2$ and $|Q| = k$. Without loss of generality, assume $|Q_1| \ge k/2$. Then, we claim that:

$$\left|\hat{\eta}_N(Q) - \bar{\eta}_N(Q)\right| \le 2\left(\frac{2\ell}{|Q_1|} + \sqrt{\frac{1}{2(|Q_1| - \ell)}\log\frac{k^2 \cdot \mathcal{N}_{\mathcal{A}}(\alpha)}{p}}\right), \tag{13}$$

where the right-hand side of Equation 13 is less than $t$ when $k$ is sufficiently large:

$$k > \frac{16\ell + 8\log k}{t^2}.$$

In particular, this holds when $Q$ is a $k$-nearest neighbor query.

Lemma 13 and Lemma E.2 bounds $\mathcal{N}_{\mathcal{A}}(\alpha) \le 8/\alpha$. So indeed, $k$ is sufficiently large:

$$\frac{16\ell + 8\log k}{t^2} \le \frac{32}{t^2}\left(\log 2 + e^2 + \log k + \log\frac{8}{\alpha} + \log\frac{1}{p}\right)$$

$$\le \frac{500}{t^2}\left(1 + \log k + \log\frac{1}{\varepsilon^{-1}(1/N)} + \log\frac{1}{p}\right) < k,$$

since $\log 2 + e^2 + \log 8 \le 12$.

Proof of claim. Let $C = \sqrt{\frac{1}{2}\log\frac{k \cdot \mathcal{N}_{\mathcal{A}}(\alpha)}{p}}$. There are two cases:

1. Case 1: $|Q_2| > 2\ell$.

$$\left|\hat{\eta}_N(Q) - \bar{\eta}_N(Q)\right| \leq \frac{1}{|Q|}\left(|Q_1| \cdot \left|\hat{\eta}_N(Q_1) - \bar{\eta}_N(Q_1)\right| + |Q_2| \cdot \left|\hat{\eta}_N(Q_2) - \bar{\eta}_N(Q_2)\right|\right)$$

$$\leq \frac{1}{|Q|}\left(4\ell + C\sqrt{\frac{|Q_1|^2}{|Q_1| - \ell}} + C\sqrt{\frac{|Q_2|^2}{|Q_2| - \ell}}\right)$$

$$\leq \frac{1}{|Q_1|}\left(4\ell + 2C\sqrt{\frac{|Q_1|^2}{|Q_1| - \ell}}\right)$$

$$= 2\left(\frac{2\ell}{|Q_1|} + \sqrt{\frac{1}{2(|Q_1| - \ell)}\log\frac{k \cdot \mathcal{N}_{\mathcal{A}}(\alpha)}{p}}\right).$$

2. Case 2: $|Q_2| \leq 2\ell$.

$$\left|\hat{\eta}_N(Q) - \bar{\eta}_N(Q)\right| \leq \frac{1}{|Q|}\left(|Q_1| \cdot \left|\hat{\eta}_N(Q_1) - \bar{\eta}_N(Q_1)\right| + |Q_2| \cdot \left|\hat{\eta}_N(Q_2) - \bar{\eta}_N(Q_2)\right|\right)$$

$$\leq \frac{1}{|Q|}\left(4\ell + \sqrt{\frac{|Q_1|^2}{2(|Q_1| - \ell)}\log\frac{k \cdot \mathcal{N}_{\mathcal{A}}(\alpha)}{p}}\right)$$

$$\leq 2\left(\frac{2\ell}{|Q_1|} + \sqrt{\frac{1}{2(|Q_1| - \ell)}\log\frac{k \cdot \mathcal{N}_{\mathcal{A}}(\alpha)}{p}}\right).$$

$\square$

*Proof of Theorem 2.* Recall that Theorem 15 proved consistency for the $k_n$-nearest neighbor rule when $\eta$ is continuous, under the assumption that almost surely no distance ties occur. That assumption only came in so that the concentration bound for $k_n$-nearest neighbor queries without ties, Corollary 2, could by applied. By replacing that with the concentration bound for $k_n$-nearest neighbor queries with tie-breaking, Lemma E.3, we immediately the result. $\square$

