# OpenReview forum: "Consistency of the $k_n$-nearest neighbor rule under adaptive sampling"
_NeurIPS.cc/2025/Conference — NeurIPS 2025 poster_

### Official Review · Reviewer_XUWz · 2025-06-30

**Clarity:** 2
**Significance:** 3
**Originality:** 3
**Rating:** 5
**Confidence:** 2

**Summary:**

In this paper, the authors consider online binary classification under an adaptive sampling model (i.e., the sequence of prediction tasks can adapt to the sequence of observed labels). A KNN based procedure is proposed and is shown to be consistent (approach Bayes optimal performance) under the assumption of "uniform absolute continuity" of the data process (a type of smoothness condition) and appropriate choice of $k$. Under a stronger assumption of ``upper doubling" on the metric space,  KNN is further shown to satisfy universal consistency for any measurable label function $\eta$. It is also established that in the worst-case setting, the $k_n$-nearest neighbor rule can fail to be consistent.

**Questions:**

- I'm a bit confused about Line 106. Are the labels not supposed to be random?

- The results seem to me very geometry-tied, which may be an artifact of the particular algorithm under consideration. Is there a sense in which kNN is the best thing to do here? The setup is very close to a bandit / online optimization setting,  where the type of guarantees / assumptions / optimal algorithms are quite different from the proposal here.

**Ethical Concerns:**

["NO or VERY MINOR ethics concerns only"]

**Final Justification:**

I believe the paper has made contributions to the problem under study and has extended previous results to new settings, although it will benefit from improvements on the presentation.

**Limitations:**

Yes it's adequately addressed.

**Paper Formatting Concerns:**

None.

**Quality:**

3

**Strengths And Weaknesses:**

Overall, the results presented in the paper constitute solid theoretical contributions, that is generalized from previous work on i.i.d to the sequential and noisy label setting. The claims are sound as far as I can tell, and the paper is well-written with clear presentation. The problem itself is also of great interest to the ML community at large.


My main concern about this paper is its practicality. The distance calculation upon seeing every new observation, and the memory requirement for such a KNN-based algorithm is not scalable to high dimension and/or large data. In another direction: how robust is the choice of $k$ here, since the smoothness parameter $\epsilon$ is almost impossible to know exactly. I also don't think conditions like upper doubling is easy to check in practice.

The paper can use some reorganization: the related work section should come earlier with a conclusion in the end. I find some parts a bit technical and better pushed to the appendix with sketches of the argument in the main text sufficient.

---

> ### Author Rebuttal · Authors · 2025-07-28
>
> Thank you for the positive feedback and the suggestions for how to improve the organization of the paper. We have already significantly updated our presentation—hopefully in a good way in terms of readability. We will continue revisions with your suggestions in mind.
>
> To address the remaining questions:
>
> **Practical considerations.**
>
> It is a good point that nearest neighbor methods are generally poorly-suited for high-dimensional data. High cost of memory and computation can be addressed using data structures like the cover tree (Beygelzimer et. al. 2006), approximate nearest neighbor search such as locality-sensitive hashing (Indyk et. al. 1997), and dataset compression (Kusner et. al. 2014). The high statistical cost coming from the curse of dimensionality is more fundamental; to overcome them requires strong priors about the learning problem.
>
> In any case, the main goal of this paper is more theoretical. We aim to illuminate the specific challenges of learning in the presence of noise in the non-worst-case online setting, show how they can be overcome in the case of this fundamental learning algorithm, and further develop the language and techniques for smoothed online learning.
>
> A positive practical consequence of this work would not necessarily be that the kn-NN algorithm is used more widely. Rather, this work shows the ways in which the worst-case setting can be extremely pathological. Overly-pessimistic algorithms that hedge unnecessarily against worst-case scenarios tend to pay the price of being sample inefficient. We hope that a downstream effect of our contribution to smoothed analysis is that it shows us how to be more optimistic, allowing us to design more sample-efficient algorithms without giving up on performance guarantees.
>
> **Choice of $k_n$.**
>
> The choice of $k_n$ is not extremely sensitive to the rate $\epsilon(\delta)$. In fact, it does not depend on it at all for our second main result, Theorem 5. For the first main result, Theorem 2, note that we do not need to know the exact rate at which $\varepsilon$ goes to zero, just an upper bound on the order. For example, consider the smoothed adversary of Haghtalab et. al. (2020), where we're guaranteed that $\varepsilon (\delta) <  L \delta$ for some constant $L > 0$. In this case, we would not need to know the constant $L$. It is enough to know that the decay rate is linear $\varepsilon = O(\delta)$.
>
> **Upper doubling condition.**
>
> An example of an upper doubling space is Euclidean space with the standard Lebesgue measure.
>
> **Confusion about Example 6.**
>
> This example is precisely aiming to illustrate the counter-intuitive behavior of non-i.i.d. data. Yes, the labels are generated randomly. However, because of how the data process decided to choose the next instance $X_n$ after observing all the past data $X_1, Y_1,\ldots, X_{n-1}, Y_{n-1}$, the dataset that one ends up with at the very end does not look random at all. In short, when data is collected adaptively, the sampling procedure itself can create patterns out of pure noise. In statistics, this is also the issue of the “garden of forking paths” (Gelman and Loken, 2013).
>
> **Geometric nature of analysis.**
>
> The arguments here are indeed very geometric, which is due to the algorithm itself being geometric in nature. The smoothed online analysis for a different learning algorithm (e.g. kernels, neural nets, gradient descent, etc.) would require different techniques. However, we do believe that the ideas and techniques we have developed here would have high transferability and utility to these research questions.
>
>
> Beygelzimer, Alina, Sham Kakade, and John Langford. "Cover trees for nearest neighbor." Proceedings of the 23rd international conference on Machine learning. 2006.
>
> Indyk, Piotr, et al. "Locality-preserving hashing in multidimensional spaces." Proceedings of the twenty-ninth annual ACM symposium on Theory of computing. 1997.
>
> Kusner, Matt, et al. "Stochastic neighbor compression." International conference on machine learning. PMLR, 2014.
>
> Gelman, Andrew, and Eric Loken. "The garden of forking paths: Why multiple comparisons can be a problem, even when there is no “fishing expedition” or “p-hacking” and the research hypothesis was posited ahead of time." Department of Statistics, Columbia University 348.1-17 (2013): 3.

---

> > ### Comment · Reviewer_XUWz · 2025-08-05
> >
> > Thanks for the clarifications -- I have no additional questions.

---

### Official Review · Reviewer_ZWJH · 2025-07-01

**Clarity:** 2
**Significance:** 3
**Originality:** 3
**Rating:** 4
**Confidence:** 3

**Summary:**

This paper studies the consistency of the k_n-nearest neighbor rule under an adaptive adversary and label noise. They show that the k_n-nearest neighbor rule is not conistent under such an adaptive adverasary. The authors complement this negative result by showing that one can recover consistency by placing additional mild assumptions on the adversary.

**Questions:**

- See Weakness above for questions regarding organization
- What are some examples of upper doubling spaces?

**Ethical Concerns:**

["NO or VERY MINOR ethics concerns only"]

**Final Justification:**

I maintain my positive score as the authors have adequately answered my questions.

**Limitations:**

yes

**Quality:**

3

**Strengths And Weaknesses:**

Strengths:
 - I think the results of the paper are interesting and provide a clear contribution to the existing literature on the consistency of k_n-nearest neighbors.
- Moreover, I think this paper contributes nicely to the growing literature on beyond-worse-case analysis
- Finally, I thought the Theorem statements were surprisingly clean and easy to parse.

Weaknesses: My main gripe with this paper is its organization. While the authors do a nice job of summarizing the main results in Section 1.2, it was not immediately clear how the remaining sections map to these main results. Moreover, I often got lost reading through some of the (often verbose) prose wondering what the main point is and how it connects to the main contributions stated in Section 1.2. For example, instead of providing a proof sketch of Proposition 7, the authors provide a "false proof of convergence." What is the point of lines 133-137?  What is the point of Section 3 how is it used in the proof of the main claims in Section 1.2?  Definition 12, Lemma 13, and Lemma 14 are not about k_n-nearest neighbor. How do these connect to the main claims and what is the point of Section 3.1? In my opinion, a paper should be focused on substantiating its main claims, with sections clearly labeled indicating which claim is being addressed. This is not the case with this paper, which to me, reads as if it were written from a stream of consciousness (e.g. lines 138-142).

---

> ### Author Rebuttal · Authors · 2025-07-28
>
> Thank you for the positive review and for pointing out where the manuscript needs copy editing. We have since improved the organization/exposition of the paper, and will continue to edit it with your comments in mind, to help align the paper’s structure to the reader’s expectation.
>
> **Verbosity.**
>
> We agree that the passages you pointed out are too meandering. They, and many others, have undergone heavy revision. Hopefully we have managed to strike a better balance between being formal/technical and human-readable.
>
> **Connection between Section 3 and main results.**
>
> To study the behavior of algorithms in the noisy but i.i.d. setting, we often rely on uniform law of large numbers (LLN) to argue that noise will tend to average out (e.g. empirical process/VC theory). But, in the worst-case online setting, the uniform LLN holds only for very restricted classes (Rakhlin et. al. 2015); noise doesn’t need to average out in the ways we might expect if we come from the i.i.d. world. This was developed in Section 2 and Appendix A, which show how the uniform LLN breaks down.
>
> However, these failures are in a sense quite pathological: under the uniform domination condition, they almost never occur, as shown in Section 3. This section is basically about the uniform law of large numbers (LLN) in the non-i.i.d. setting for specific function classes. This is later used to control the averaged behavior of noisy labels: does the majority vote over the labels of the kn-nearest neighbors behave “as expected” (at least coming from the i.i.d. perspective)?
>
> The reason that Section 3 seems so decoupled from the main results about kn-NN is because these techniques are likely relevant for more general smoothed analysis for adaptive/online learning, and so we had extracted them out. We agree that re-organization and improved sign-posting will better help the logical flow of the paper.
>
> **Examples of upper doubling spaces.**
>
> The upper doubling space is a relaxation of the doubling measure space. This includes Euclidean space with Lebesgue measure. More generally, any doubling metric space with the Hausdorff measure is upper doubling. These are all standard metric measure spaces used in geometric measure theory and metric-based learning theory.

---

> > ### Comment · Reviewer_ZWJH · 2025-08-01
> >
> > I thank the authors for their response, and maintain my positive score.

---

### Official Review · Reviewer_1qFD · 2025-07-03

**Clarity:** 2
**Significance:** 2
**Originality:** 2
**Rating:** 3
**Confidence:** 3

**Summary:**

The paper considers the problem of asymptotically proper prediction of binary labels for online (adversarial) sampling from a given space. The authors provide a complex mathematical analysis of the algorithm based on $k_n$-Nearest Neighbor, for certain parameters $k_n$ depending on the sampling round n and for data generation process that, although allowing adversarial actions, it guarantees that intractable worst-cases generation patterns occur with negligible probability. Additionally, the authors claim that their analysis includes the presence of noise.

**Questions:**

1.	Compare results with previous work, focusing on differences.
2.	Define noise formally.
3.	Emphasize novel techniques and their impact in the analysis.
4.	How to obtain proofs of Theorems 2 and 5 from technical lemmas?

**Ethical Concerns:**

["NO or VERY MINOR ethics concerns only"]

**Final Justification:**

The rebuttal helped me to understand better the significance of the presented results vs previous work, which is above acceptance bar. However, I still do not see substantial technical novelty, which seem to me a kind of "randomized relaxation" (this is what I meant by Yao principle - the process gets more randomness, and Yao principle connects random executions with worst case deterministic ones) of the analysis in the previous work(s). I may be wrong in it, but the authors did not convince me otherwise with their too general arguments. I keep my borderline reject score; however I do not oppose acceptance if other reviewers are convinced that the paper is above NeurIPS acceptance bar.

**Limitations:**

yes

**Quality:**

3

**Strengths And Weaknesses:**

Pros:

-	The problem itself is among fundamental theoretical ML problems

-	Mathematical analysis is quite complex

Cons:

-	The novelty of the analyzed setting is unclear, comparing to the previous work. It looks like the main difference is noise, but I couldn’t find its formal definition and use in technical parts. I also couldn’t find a satisfactory comparison with the previous work(s)

-	The result holds under certain assumptions, including Borel measurable spaces, continuous label distribution function (or measurable function in upper doubling space), etc.

-	The authors assume that the reader is fluent in various aspects of mathematical statistics and topology

Detail comments:

The title does not say anything about the actual angles of the paper – the problem, adversarial setting, noise, etc.

The concept of noise is not formally defined in the main paper, only by the means of examples. It is not clear what are real difference in the setting and techniques between this work and Dasgupta and So, NeurIPS 2024. The authors are not clear in referencing results to previous work, especially results stated without any proof.

The proof of the main Theorem 5 is somehow stopped and hand-waved closer to the end. I could follow the analysis (up to some presented level of details) up to Lemma 17, but Section 5 is far too sketchy.

---

> ### Author Rebuttal · Authors · 2025-07-28
>
> Thank you for taking the time to review and providing a lot of detail as to where the exposition and relationship to prior work can be improved. To address your questions:
>
> **Relationship to prior work.**
>
> One nice way to place results in learning theory is along two dimensions of “data complexity” and “model complexity”. Data complexity refers to how complicated the process generating the data is: from i.i.d. data, to worst-case data from an adaptive adversary. Model complexity refers to how complex the relationship between instances and labels can be: from parametric (Littlestone, VC, etc.) classes, to over-parametrized/non-parametric classes (neural nets, kernel methods, distance-based learning).
>
> The vast majority of work in learning theory considers two extremes (such as non-parametric learning in i.i.d. settings, or online learning with Littlestone classes), leaving the non-worst-case region in between fairly uncharted. There are a few exceptions.
>
> The line of work from Haghtalab et. al. (2020) considers parametric learning with smoothed adversaries, and they are interested in algorithms that attain finite-sample generalization bounds. The line of work from Hanneke et. al. (2021) is about understanding the Pareto frontier of learnability in the limit with respect to data/model complexity. Correspondingly, the types of data processes they study complement their goals: the former impose stronger, quantitative conditions, while the latter impose minimal, measure-theoretic assumptions. These two lines of work were connected by Dasgupta and So, who focused on the 1-NN learning rule in the non-parametric regime and proposed the general class of dominated adversaries interpolating the two (see also their related works).
>
> However, as they restricted themselves to the realizable/noiseless setting, they left open the problem of adaptivity in the presence of noise. Can an adaptive adversary take advantage of noisy labels (and how)? Are stronger constraints on the data process needed for learning? In this work, we illustrate the challenges of online learning in the presence of noise, which do not seem reducible to the noiseless setting. Indeed, new techniques were needed to analyze the noisy setting. Interestingly, we did not need to introduce additional constraints on the data processes (uniform domination was enough), and the standard kn-nearest neighbor rule works out-of-the-box for far more general settings than previously known.
>
> Thus, this work is also of interest in its own right to the non-parametric estimation community, which has classically studied the i.i.d. consistency of nearest neighbor methods.
>
> **Clarification on noise.**
>
> Dasgupta and So (2024) considered the realizable/noiseless setting where the labels are given by a deterministic function of the instances, $y = f(x)$ where $f : \mathcal{X} \to$ {0,1}. In this work, noise just means that the labels can now be random. Our setting is formally defined in the first paragraph (line 13-20). Using our language, the setting considered by Dasgupta and So corresponds to the case where the conditional mean $\eta(x)$ is either $0$ or $1$ for each $x \in\mathcal{X}$.
>
> Perhaps surprisingly, the existence of label noise introduces a lot of new challenges to analysis (see Sections 2 and 3), which we discuss next.
>
> **New techniques and impact.**
>
> To study the behavior of algorithms in the noisy but i.i.d. setting, we often rely on uniform law of large numbers (LLN) to argue that noise will tend to average out (e.g. empirical process/VC theory). But, in the worst-case online setting, the uniform LLN holds only for very restricted classes (Rakhlin et. al. 2015); noise doesn’t need to average out in the ways we might expect if we come from the i.i.d. world. We develop this in Section 2 and Appendix A, which show how the uniform LLN breaks down.
>
> However, we also argue that these failures are in a sense quite pathological: under the uniform domination condition, they almost never occur, as shown in Section 3. We’ve decoupled these techniques from the main results for kn-NN, since they are likely relevant for more general smoothed analysis for adaptive/online learning.
>
> There were other technical novelties that may be of more specific interest depending on the researcher (e.g. Lemma 17 - coupling technique for analyzing dominated processes, Lemma 18 - reduction to smoothed processes, Lemma D.1 - control over long-term kn-NN influence, Lemma D.4 - control over distances to the kn-NN).
>
> **Improved proof sketches.**
>
> Thank you for pointing out where we should bolster our proof sketches. We will also make sure to include more details in some of our proofs (especially in Theorem 5). Regarding Theorem 5, the main proof can be summarized with the following steps
>
> 1. We observe that our data sequence can be thought of as a sequence generated from a Lipschitz dominated adversary combined with an arbitrary adversary (with the arbitrary adversary only outputting during a small fraction of the times).
>
> 2. We then note that $k_n$ nearest neighbors run purely over the Lipschitz adversary immediately satisfies consistency due to our prior results about consistency. In particular, this is a case where a regular sequence $(k_n)_n$ will satisfy the conditions needed for consistency outlined above.
>
> 3. We bound how often the points from the arbitrary adversary “pollute” the nearest neighbors of our sequence (thereby impacting the consistency in 2). The key idea here is to note that consistency in 2 holds with a margin for almost all points (i.e., the fraction of nearest neighbors that are correctly labeled is above 1/2 with a margin), and thus changing the output of nearest neighbors requires a fraction of the neighbors actually coming from the arbitrary adversary.
>
> 4. We bound the number of instances where a fraction of its nearest neighbors come from the arbitrary adversary. To do so, we apply the techniques from Dasgupta and So (2024) that they applied for their long-term influence bound. The main technical challenge here is to extend a bound on the frequency that a set of points forms a 1-NN to a bound on the frequency of a set of points comprise a fraction of points of the $k_n$ nearest neighbors. This is resolved through technical Lemma 19.

---

> > ### Comment · Reviewer_1qFD · 2025-08-08
> >
> > Thank you for thorough explanation, I have a better understanding of your work. However, my slight reservations about assumptions and how technically novel your analysis is comparing to Dasgupta and So NeurIPS 2024 (I understand that the noise makes difference, but one could still use Yao technique to "randomize" parameterized deterministic techniques - unfortunately, I have not been familiar with that work prior reading your paper). I keep my mildly positive score for now.

---

> > > ### Author Response · Authors · 2025-08-08
> > >
> > > Thanks for your response. The main difference in our setting and the one considered by Dasgupta and So (2024) is this: can the learner receive wrong labels? The earlier work: no. Our work: yes.
> > >
> > > There are many learning settings where the presence of noise makes things much harder, and so the assumption of realizability is a significant limitation of the prior work. It is definitely not clear *a priori* what to do in the online setting with noise. Our Example 6 and Proposition 7 are counter-examples showing what makes noise hard; these examples cannot happen in the realizable setting.
> > >
> > > But what specifically would you like to see in order to raise your score from a 3-borderline reject to perhaps a mildy more positive score? We would like to address your concerns, as we believe this work is a worthwhile follow up and of interest to multiple communities. It rounds out our understanding about classic nearest neighbor methods in the smoothed online setting.
> > >
> > >  &nbsp;
> > >
> > > P.S. On Yao's principle: the problem isn't really about randomized versus deterministic algorithms here, and it is not clear that it can be used to extend the analysis about 1-NN from Dasgupta and So, to the kn-NN algorithm here. In any case, the kn-NN algorithm is not a randomized algorithm (well, almost surely except for tie-breaking), where all sources of randomness comes from the data stream.

---

### Official Review · Reviewer_hNva · 2025-07-03

**Clarity:** 3
**Significance:** 3
**Originality:** 3
**Rating:** 5
**Confidence:** 3

**Summary:**

This paper studies the consistency of $k_n$-nn classifiers under noise & adaptive (non-iid) data generation processes in general metric spaces. This continues a line of work on the consistency of the $k_n$-classifier under more challenging settings and belongs to the broader scope of online learning in between worst-case adversarial and iid . In particular this paper extends the results of Dasgupta and So (NeurIPS 2024) from 1-NN to general $k_n$-nn and from noiseless to benign noise. They have two main results:  $k_n$ is consistent (for specific choices of $(k_n)_n$) under a distributional assumption related to absolutely continuity if additionally

1.  and the Bayes optimal classifier is continuous, or
2. the metric space satisfies a certain doubling dimension based assumption.

They also discuss that under weaker assumptions consistency typically fails.

**Questions:**

Consider adding some corollaries for common settings (Euclidean etc.).

Your assumption on $k_n$ in Equation (2) requires knowledge of $\varepsilon$. It is of course not really realistic to know $\varepsilon$. Is there some way to make it work without knowing $\varepsilon$? E.g., guess it by some doubling trick etc.

**Ethical Concerns:**

["NO or VERY MINOR ethics concerns only"]

**Final Justification:**

Good paper and important problem.

**Limitations:**

yes

**Paper Formatting Concerns:**

.

**Quality:**

3

**Strengths And Weaknesses:**

This paper studies an interesting and challenging problem. It greatly extends the work by Dasgupta and So (2024) on consistency results for 1-nn to $k_n$-nn. It fits nicely into the recent line of work on online learnability between adversarial/worst-case and iid (i.e., learning under general adaptive stochastic processes).

The structure of the paper is not ideal. On first sight the purpose of sections 2-5 is not very clear. Section 2 and 3 provide some more intuition on the challenges in this setting, Section 4 and 5 give a proof overview on the two main results. Perhaps try to clarify this.

Also the discussion of related work is rather limited. Please discuss more exactly how you relate to similar settings (e.g., Hanneke et al., 2021). In particular please discuss how your assumption relate to previous assumptions in other works (as you did for Haghtalab et al., saying you generalize it).

---

> ### Author Rebuttal · Authors · 2025-07-27
>
> Thank you for the careful review and positive feedback. We agree with all of your suggestions. We have since improved the organization/exposition of the paper, and we will deepen the related works section. To briefly address it here:
>
> **Relationship to prior work.**
>
>
> One nice way to place results in learning theory is along two dimensions of “data complexity” and “model complexity”. Data complexity refers to how complicated the process generating the data is: from i.i.d. data, to worst-case data from an adaptive adversary. Model complexity refers to how complex the relationship between instances and labels can be: from parametric (Littlestone, VC, etc.) classes, to over-parametrized/non-parametric classes (neural nets, kernel methods, distance-based learning).
>
> The vast majority of work in learning theory considers two extremes (such as non-parametric learning in i.i.d. settings, or online learning with Littlestone classes), leaving the non-worst-case region in between fairly uncharted. There are a few exceptions.
>
> The line of work from Haghtalab et. al. (2020) considers parametric learning with smoothed adversaries, and they are interested in algorithms that attain finite-sample generalization bounds. The line of work from Hanneke et. al. (2021) is about understanding the Pareto frontier of learnability in the limit with respect to data/model complexity. Correspondingly, the types of data processes they study complement their goals: the former impose stronger, quantitative conditions, while the latter impose minimal, measure-theoretic assumptions. These two lines of work were connected by Dasgupta and So, who focused on the 1-NN learning rule in the non-parametric regime and proposed the general class of dominated adversaries interpolating the two (see also their related works).
>
> However, as they restricted themselves to the realizable/noiseless setting, they left open the problem of adaptivity in the presence of noise. Can an adaptive adversary take advantage of noisy labels (and how)? Are stronger constraints on the data process needed for learning? In this work, we illustrate the challenges of online learning in the presence of noise, which do not seem reducible to the noiseless setting. Indeed, new techniques were needed to analyze the noisy setting. Interestingly, we did not need to introduce additional constraints on the data processes (uniform domination was enough), and the standard kn-nearest neighbor rule works out-of-the-box for far more general settings than previously known.
>
> Thus, this work is also of interest in its own right to the non-parametric estimation community, which has classically studied the i.i.d. consistency of nearest neighbor methods.
>
> **Response to question.**
>
> On the question of whether we can remove the need to know $\varepsilon$ for the first result Theorem 2. This is an interesting question, which is in part what led to the second result Theorem 5. The latter result does not require rate assumptions on $\varepsilon$, but it achieves this by making use of stronger, but standard, metric assumptions (upper doubling spaces, such as Euclidean space).
>
> For Theorem 2 specifically, note that we do not need to know the exact rate at which $\varepsilon$ goes to zero, just an upper bound on the order. For example, consider the smoothed adversary of Haghtalab et. al. (2020), where we're guaranteed that $\varepsilon (\delta) <  L \delta$ for some constant $L > 0$. In this case, we would not need to know the constant $L$. It is enough to know that the decay rate is linear $\varepsilon = O(\delta)$. It would be interesting if we could apply the doubling trick to the decay rate. Although, this would come at the cost of imposing additional restrictions on the sequence $(k_n)_n$ for which the result would apply.

---

### Decision · Program_Chairs · 2025-09-17

**Decision:**

Accept (poster)

**Comment:**

Summary
This paper considers the adaptive sampling model of online learning, specifically, the k-nearest neighbor learner. Under mild assumption that the process generating the instances is uniformly absolutely continuous and that the underlying conditional label distribution is continuous, this paper shows that k-nearest neighbor rule is consistent.

Strengths
- This paper studies an interesting and challenging problem, greatly extending Dasgupta and So (2024) on consistency results for 1-nn to k_n-nn.
- This paper conteibutes nicely to the growing literature on beyond-worse-case analysis.
- The problem studied is a fundamental theoretical problem.

Weaknesses
- Multiple reviewers mentioned that the structure of the paper could be improved to be more easily readable.
- The discussion of related work could be improved.
- Reviewer XUWz concerned the memory requirement for this KNN algorithm might limit the scalability, and the robustness of the choice of k.